# Imaging of the pial arterial vasculature of the human brain in vivo using high-resolution 7T time-of-flight angiography

Saskia Bollmann[1,2,3]*, Hendrik Mattern[4], Michaël Bernier[1,2], Simon D Robinson[3,5,6,7], Daniel Park[1], Oliver Speck[4,8,9,10], Jonathan R Polimeni[1,2,11]

[1]Athinoula A. Martinos Center for Biomedical Imaging, Massachusetts General Hospital, Charlestown, United States; [2]Department of Radiology, Harvard Medical School, Boston, United States; [3]Centre for Advanced Imaging, The University of Queensland, Brisbane, Australia; [4]Department of Biomedical Magnetic Resonance, Institute of Experimental Physics, Otto-von- Guericke-University, Magdeburg, Germany; [5]High Field MR Centre, Department of Biomedical Imaging and Image-guided Therapy, Medical University of Vienna, Vienna, Austria; [6]Karl Landsteiner Institute for Clinical Molecular MR in Musculoskeletal Imaging, Vienna, Austria; [7]Department of Neurology, Medical University of Graz, Graz, Austria; [8]German Center for Neurodegenerative Diseases, Magdeburg, Germany; [9]Center for Behavioral Brain Sciences, Magdeburg, Germany; [10]Leibniz Institute for Neurobiology, Magdeburg, Germany; [11]Division of Health Sciences and Technology, Massachusetts Institute of Technology, Cambridge, United States

*For correspondence: saskia.bollmann@cai.uq.edu.au

**Competing interest:** The authors declare that no competing interests exist.

**Abstract** The pial arterial vasculature of the human brain is the only blood supply to the neocortex, but quantitative data on the morphology and topology of these mesoscopic arteries (diameter 50–300 μm) remains scarce. Because it is commonly assumed that blood flow velocities in these vessels are prohibitively slow, non-invasive time-of-flight magnetic resonance angiography (TOF-MRA)—which is well suited to high 3D imaging resolutions—has not been applied to imaging the pial arteries. Here, we provide a theoretical framework that outlines how TOF-MRA can visualize small pial arteries in vivo, by employing extremely small voxels at the size of individual vessels. We then provide evidence for this theory by imaging the pial arteries at 140 μm isotropic resolution using a 7 Tesla (T) magnetic resonance imaging (MRI) scanner and prospective motion correction, and show that pial arteries one voxel width in diameter can be detected. We conclude that imaging pial arteries is not limited by slow blood flow, but instead by achievable image resolution. This study represents the first targeted, comprehensive account of imaging pial arteries in vivo in the human brain. This ultra-high-resolution angiography will enable the characterization of pial vascular anatomy across the brain to investigate patterns of blood supply and relationships between vascular and functional architecture.

## Editor's evaluation

This article revisits a classic magnetic resonance imaging technique to image brain vasculature, showing that small blood vessels, deemed too difficult to image, could be targeted effectively with extremely high-resolution imaging. This proof-of-concept, both theoretical and experimental, may be of use for building detailed models of brain physiology and detecting fine alterations of blood vessels in disease.

## Introduction

The pial arterial vasculature of the cerebrum consists of smaller distal arteries and arterioles that cover the cerebral cortical surface, and connects the branches of the three major supplying arteries of the cerebrum—the anterior, middle, and posterior cerebral arteries—with the penetrating intracortical arterioles, which deliver blood into the cortical grey matter (*Cipolla, 2009*; *Jones, 1970*). Notably, the pial arterial vasculature is the only source of blood supply to the neocortex (*Mchedlishvili and Kuridze, 1984*), and its redundancies are organized in three tiers: global re-routing of blood through the circle of Willis, intermediate anastomoses between branches originating from the three major arteries, and local loops formed by pial anastomoses within the same branch (*Blinder et al., 2010*; *Duvernoy et al., 1981*).

Although extensive anatomical studies have described topological properties and the relevant constituents of the pial arterial vasculature (*Pfeifer, 1930*; *Szikla et al., 1977*), *quantitative* data of the human pial arterial vasculature remain scarce (*Cassot et al., 2006*; *Helthuis et al., 2019*; *Hirsch et al., 2012*; *Payne, 2017*; *Schmid et al., 2019*). The by far still 'most comprehensive and influential work' (*Hirsch et al., 2012*) is the detailed description of the pial vasculature by *Duvernoy et al., 1981*, which examined 25 brains using intravascular ink injections. As indispensable as this dataset has been, 3D reconstructions of the vascular network and surrounding anatomy were not provided in this study. A second recent analysis performed by *Helthuis et al., 2019*, used corrosion casts from four brain specimens and provided valuable insights into the branching pattern of the arterial vasculature. However, only limited information can be obtained in this way about the morphometry of vessels, in particular their position and geometric relationship with the cortex. Further, the elaborate preparation and the limitation to ex vivo samples restrict the applicability of these methods when, for example, one wants to characterize the large variability across individuals in pial arterial morphometry (*Alpers et al., 1959*; *Beevor and Ferrier, 1997*; *Cilliers and Page, 2017*; *Gomes et al., 1986*; *Papantchev et al., 2013*; *Stefani et al., 2000*; *van der Zwan et al., 1993*) and function (*Baumbach and Heistad, 1985*; *van Laar et al., 2006*).

Given the central role of the pial arterial vasculature for healthy human brain function (*Hirsch et al., 2012*; *Iliff et al., 2013*; *Xie et al., 2013*), its impact on functional magnetic resonance imaging (fMRI) signals (*Bright et al., 2020*; *Chen et al., 2020*), and its involvement in numerous cerebrovascular diseases (*Hetts et al., 2017*; *McConnell et al., 2016*), there is a clear need to be able to image the pial arterial vasculature in individual subjects to characterize its morphometry and topology including arterial diameter and geometric relationship with the cortex (*Mut et al., 2014*). For example, many intracranial pathologies have selective involvement of superficial vessels (*Ginat et al., 2013*; *Herz et al., 1975*; *Hetts et al., 2017*; *McConnell et al., 2016*; *Song et al., 2010*; *Uhl et al., 2003*), and the outcome of stroke patients is heavily dependent on the status of the collateral circulation (*Ginsberg, 2018*; *Raymond and Schaefer, 2017*). Yet, existing hemodynamic modelling approaches have to synthesize the pial arterial vasculature either implicitly (*Park et al., 2020*) or explicitly (*Ii et al., 2020*) from an incomplete quantitative understanding of its morphometry and variability. In addition, modelling of hemodynamic changes in response to neural activity needs to account for the effect of pial vessels (*Markuerkiaga et al., 2016*; *Polimeni et al., 2010*; *Uludağ and Blinder, 2018*). Although signal changes in fMRI data using a blood-oxygenation-level-dependent (BOLD) contrast arise predominantly on the venous side of the vascular hierarchy (*Ogawa et al., 1990*), the strongest *vascular* response to neural activity is located in intracortical arteries and arterioles (*Hillman et al., 2007*; *Vanzetta et al., 2005*), and significant diameter changes in upstream arteries have been observed (*Bizeau et al., 2018*; *Cho et al., 2012*; *Cho et al., 2008*). With the recent interest in cerebral blood volume-based fMRI (*Huber et al., 2014*) and the numerous accounts of vascular contributions found in BOLD fMRI signals (*Amemiya et al., 2020*; *Bright et al., 2020*; *Chen et al., 2020*; *Drew et al., 2020*), including a detailed, subject-specific depiction of the underlying angio-architecture, would immensely augment forthcoming modelling initiatives (*Havlicek and Uludağ, 2020*; *Tak et al., 2014*).

So far, numerous approaches exist to image large, *macro*scopic proximal intracranial arteries such as the circle of Willis including its branches and the basal arteries with magnetic resonance imaging (MRI) (*Carr and Carroll, 2012*). Similarly, the density of *micro*scopic, parenchymal vessels can be quantified through techniques such as vessel size imaging and capillary density imaging (*Kiselev et al., 2005*; *Troprès et al., 2001*; *Xu et al., 2011*), and newer methods for estimates of cerebral

blood volume of the arteriolar side of the microvascular tree (*Hua et al., 2019*). However, direct non-invasive imaging of the *meso*scopic vessels of the pial arterial network, that is, arteries with a diameter between 50 and 300 μm, is not in current use, either in clinical practice or in basic research.

Recent studies have shown the potential of time-of-flight (TOF)-based magnetic resonance angiography (MRA) at 7 Tesla (T) in subcortical areas (*Bouvy et al., 2016*; *Bouvy et al., 2014*; *Ladd, 2007*; *Mattern et al., 2018*; *Schulz et al., 2016*; *Von Morze et al., 2007*). In brief, TOF-MRA uses the high signal intensity caused by inflowing water protons in the blood to generate contrast, rather than an exogenous contrast agent. By adjusting the imaging parameters of a gradient-recalled echo (GRE) sequence, namely the repetition time ($T_R$) and flip angle, the signal from static tissue in the background can be suppressed, and high image intensities are only present in blood vessels freshly filled with non-saturated inflowing blood. As the blood flows through the vasculature within the imaging volume, its signal intensity slowly decreases. (For a comprehensive introduction to the principles of MRA, see for example *Carr and Carroll, 2012*.) At ultra-high field, the increased signal-to-noise ratio (SNR), the longer $T_1$ relaxation times of blood and grey matter, and the potential for higher resolution are key benefits (*Von Morze et al., 2007*). However, the current description of the magnetic resonance physics underlying TOF-MRA (*Brown et al., 2014a*; *Carr and Carroll, 2012*) is not tailored to imaging pial arteries, because their small diameter and complex branching pattern require particular considerations. For example, while the advantage of reduced voxel size for imaging small arteries has empirically been shown numerous times (*Haacke et al., 1990*; *Mattern et al., 2018*; *Von Morze et al., 2007*), the sentiment prevails (*Chen et al., 2018*; *Masaryk et al., 1989*; *Mut et al., 2014*; *Park et al., 2020*; *Parker et al., 1991*; *Wilms et al., 2001*; *Wright et al., 2013*) that the slower flow in small arteries should significantly diminish the TOF effect (*Haacke et al., 1990*; *Pipe, 2001*), and perhaps for that reason imaging the pial arterial vasculature in vivo has received little attention. Here, we revisit the topic of high-resolution TOF-MRA to investigate the feasibility of imaging the pial arterial vasculature in vivo at 7 T. We demonstrate, based on simulations and empirical data, that pial arteries can be detected reliably. Note that while some, particularly larger pial arterials, have undoubtedly been

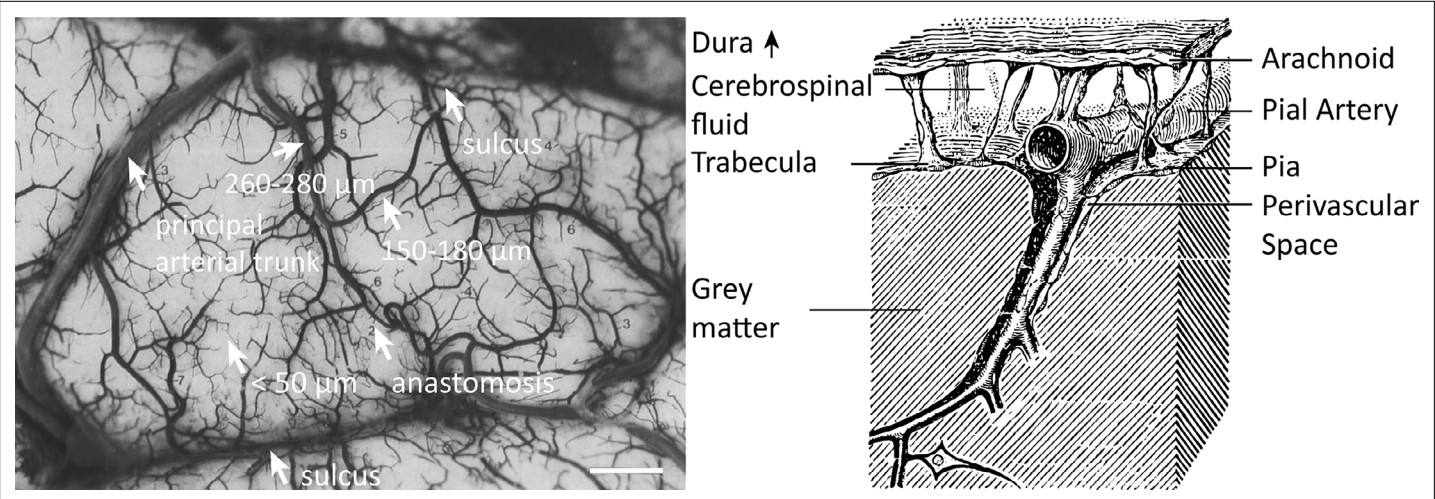

**Figure 1.** Properties of the pial arterial vasculature of the human brain.
Left: The pial vascular network on the medial orbital gyrus depicted using intravascular India ink injection (adapted with permission from *Duvernoy, 1999*; this is not covered by the CC-BY 4.0 license and further reproduction of this panel would need permission from the copyright holder). The arrows indicate the average diameter of central pial arteries (260–280 μm), peripheral pial arteries (150–180 μm), and pial arterioles (<50 μm). Pial anastomoses are commonly formed by arteries ranging from 25 to 90 μm in diameter. For reference, the diameter of intracortical penetrating arterioles is approximately 40 μm (scale bar: 2.3 mm). Right: Pia-arachnoid architecture (adapted with permission from *Ranson and Clark, 1959*; this is not covered by the CC-BY 4.0 license and further reproduction of this panel would need permission from the copyright holder) illustrates the complex embedding of pial vessels, which are surrounded by various membrane layers that form the blood-brain barrier, cerebrospinal fluid, and grey matter (*Marin-Padilla, 2012*; *Mastorakos and McGavern, 2019*; *Ranson and Clark, 1959*).

visualized incidentally in high-resolution images, we present here the first targeted, comprehensive account of imaging the pial arteries.

In the following, we first summarize the relevant properties of the pial arterial vasculature, in particular vessel diameter, blood velocity, and blood delivery time. With this in mind, we revisit the theory behind 3D TOF-MRA and derive optimal parameters for imaging these vessels. We then demonstrate the supra-linear relationship between vessel contrast and voxel size and explore the requirements for accurate vessel detection. This work argues that, from a physiological and theoretical perspective, it is indeed possible to image the pial arterial network using a TOF-based contrast. Subsequently, we present results from several experiments supporting our hypothesis and ultimately employ high-resolution TOF-MRA with prospective motion correction to image the pial arterial vasculature at 140 μm isotropic resolution.

## Theory
### Anatomical architecture of the pial arterial vasculature

The pial arterial vasculature consists of smaller arteries and arterioles on the cortical surface (*Figure 1*, left) (*Duvernoy, 1999*). Pial arteries range in diameter from 280 to 50 μm and connect to the smaller (<50 μm) intracortical arterioles. Similar values of 700–30 μm in pial arterial diameter were also reported by *Nonaka et al., 2003b*, who exclusively characterized the calcarine fissure. The recent work on the topology of the vascular system by *Hirsch et al., 2012*, might create the impression that mesoscopic pial vessels entirely cover the cortical surface (e.g. Figure 2 in *Hirsch et al., 2012*). However, the segmentation provided by *Duvernoy, 1999*, shows that pial cerebral blood vessels are rather sparse, and only approximately 17% of the human cortex is covered by arteries and 11% by veins. (To obtain a rough estimate of their density, we segmented the stylized drawing accompanying *Figure 1* (*Duvernoy, 1999*); see also subsection *Data analysis* below.) Pial arteries exhibit a distinctive branching pattern on the cortical surface, in that an arterial branch arises at a nearly right angle from its parent vessel (*Rowbotham and Little, 1965*), usually with a significantly reduced diameter. In the absence of ground-truth data, this conspicuous feature constitutes an important prior that can be used as an image quality marker: simply, the more right-angled branches are detected, the higher the sensitivity to small arteries.

From a cross-sectional point of view cutting through the cortex, pial arteries are embedded in the meningeal layers between the pia and arachnoid mater, that is, the subarachnoid space (*Figure 1*, right). Thus, their immediately surrounding tissue is not only grey matter, but a mixture of grey matter, cerebrospinal and interstitial fluid, and meningeal and glial tissue (*Marin-Padilla, 2012*). In the next section, we will propose a simplified partial-volume model to account for the small diameter of pial arteries. Nevertheless, one should keep in mind that the actual surrounding of pial arteries consists of a number of different tissue classes and fluids, with considerable differences in $T_1$ values, especially between grey matter, arterial blood, and cerebrospinal fluid (*Rooney et al., 2007*; *Wright et al., 2008*).

Along with their diameter, the blood velocity within pial arteries is an important factor determining the TOF contrast. However, blood velocity values in human pial arteries are difficult to obtain due to their small size. Using optical imaging in cats, *Kobari et al., 1984*, measured blood velocities in pial arteries ranging from 20 to 200 μm diameter. For example, in pial arteries with a diameter between 100 and 150 μm, the centreline velocity was 42.1 mm/s. Similar values were obtained by *Nagaoka and Yoshida, 2006*, for arterioles in the human retina using laser Doppler velocimetry, where the average centreline velocity for first-order arterioles with an average diameter of 107.9 μm was 41.1 mm/s. The only study in humans measuring blood velocity in small cerebral arteries was performed by *Bouvy et al., 2016*, using quantitative flow MRI. They estimated mean blood velocities of 39–51 mm/s in the basal ganglia, where arterial diameters range from 175 to 668 μm (*Djulejić et al., 2015*). *Figure 2* provides a comprehensive overview of all blood velocity values and vessel diameters reported for pial arteries. Note that optical imaging methods (*Kobari et al., 1984*; *Nagaoka and Yoshida, 2006*) are expected to considerably overestimate the mean velocity within the vessel, whereas MRI-based measurements can severely underestimate the velocity values due to partial-volume effects (*Bouvy et al., 2016*; *Hofman et al., 1995*; *Tang et al., 1993*). This might explain the large discrepancies that are apparent in *Figure 2* between optical and MRI-based methods. Due to the gradual reduction in vessel diameter along the vascular tree, the reported velocity values only apply to the very last branch,

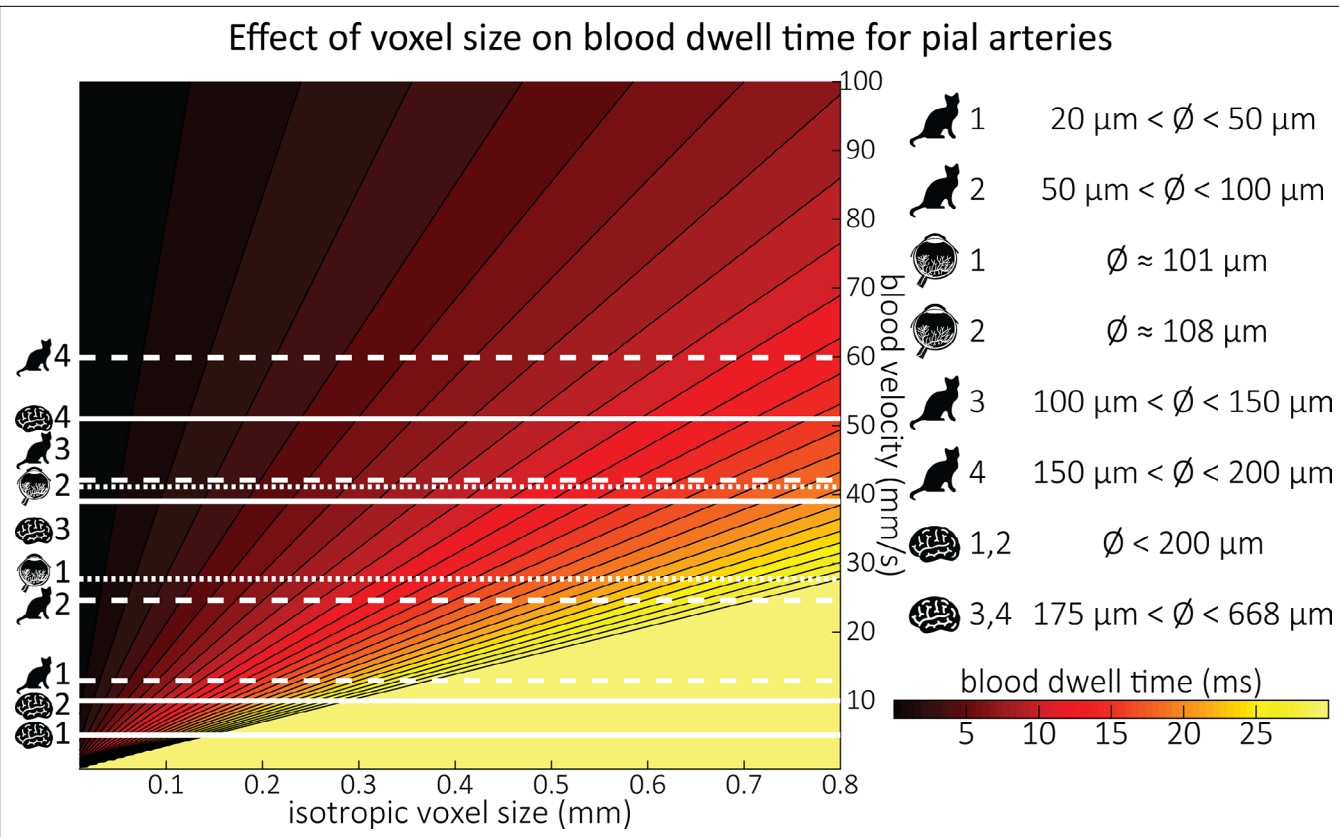

**Figure 2.** Blood dwell time (ms) as a function of blood velocity and voxel size. For small voxel sizes, blood dwell times are short even for low blood velocities. The horizontal white lines indicate blood velocities reported in humans (solid lines) in the centrum semiovale (1 and 2) and the basal ganglia (3 and 4) (*Bouvy et al., 2016*), in cats (dashed lines) for various pial artery diameters (*Kobari et al., 1984*), and human retina (dotted lines) (*Nagaoka and Yoshida, 2006*). For reference, red blood cells in the capillaries are estimated to move at approximately 1 mm/s (*Cipolla, 2009*; *Wei et al., 1993*), whereas blood velocities in the major cerebral arteries can reach 1000 mm/s (*Lee et al., 1997*; *Molinari et al., 2006*).

and faster velocities are expected along much of the vasculature. For example, blood velocities in the three major cerebral arteries exceed the velocity values depicted here by at least an order of magnitude (*Lee et al., 1997*; *Molinari et al., 2006*).

The *blood delivery time* for a given voxel containing a vessel is the time it takes for the blood water spins to reach that voxel after they first enter the imaging slab, and is a crucial imaging parameter when estimating blood signal enhancement in TOF-MRA. To directly compute blood delivery time from blood velocity, one would need a detailed velocity profile of the blood and its path along the arterial tree together with the exact slab prescription for each scan. While this is not feasible, we can find an upper bound on the blood delivery time expected for pial arteries using estimates of arterial transit time used in arterial spin labelling (*Detre et al., 1992*). The *arterial transit time*, which includes the transport of blood water spins from the large carotid arteries in the neck through the pial arterial vasculature and the capillaries into the tissue, is estimated to be between 500 and 1500 ms in healthy grey matter (*Alsop et al., 2015*). Since the coverage or slab thickness in TOF-MRA is usually kept small to minimize blood delivery time by shortening the path length of the vessel contained within the slab (*Parker et al., 1991*), and because we are focused here on the pial vasculature, we have limited our considerations to a maximum blood delivery time of 1000 ms, with values of few hundreds of milliseconds being more likely.

In summary, targeting pial arteries places us into an imaging regime of mesoscopic vessels (50–300 µm diameter) in a complex branching pattern on a folded surface with blood velocities of 10–50 mm/s and blood delivery times of 200–700 ms. When revisiting the theory behind TOF-MRA for mesoscopic pial arteries, whose diameters are at the size of the imaging voxels or smaller, we thus need to consider partial-volume effects, and take into account that: (i) there is no preferential vessel

orientation; (ii) while the blood delivery times are long relative to macroscopic arteries, they are still sufficiently short that the blood arrives in the pial arteries in a small fraction of the overall acquisition time; and (iii) the blood passes swiftly through each voxel (i.e. in a time which is short relative to commonly used repetition times).

## Flow-related enhancement

Before discussing the effects of vessel size, we briefly revisit the fundamental theory of the flow-related enhancement (FRE) effect used in TOF-MRA. Taking into account the specific properties of pial arteries, we will then extend the classical description to this new regime. In general, TOF-MRA creates high signal intensities in arteries using inflowing blood as an endogenous contrast agent. The object magnetization—created through the interaction between the quantum mechanical spins of water protons and the magnetic field—provides the signal source (or magnetization) accessed via excitation with radiofrequency (RF) waves (called RF pulses) and the reception of 'echo' signals emitted by the sample around the same frequency. The $T_1$ contrast in TOF-MRA is based on the difference in the steady-state magnetization of static tissue, which is continuously saturated by RF pulses during the imaging, and the increased or enhanced longitudinal magnetization of inflowing blood water spins, which have experienced no or few RF pulses. In other words, in TOF-MRA, we see enhancement for blood that flows into the imaging volume. The steady-state longitudinal magnetization $M_{zS}^{\text{tissue}}$ of static tissue imaged with a (spoiled) FLASH sequence can be calculated (**Brown et al., 2014a**) using the Ernst equation:

$$M_{zS}^{\text{tissue}} = \frac{M_0 \left(1 - e^{-T_R/T_1^{\text{tissue}}}\right)}{1 - e^{-T_R/T_1^{\text{tissue}}} \cdot \cos\theta}, \tag{1}$$

with $M_0$ being the thermal equilibrium magnetization before any RF pulses, $T_R$ the repetition time, $\theta$ the excitation flip angle, and $T_1^{\text{tissue}}$ the longitudinal relaxation time of the tissue.

To generate a TOF contrast, a higher flip angle than the optimal Ernst angle (**Ernst and Anderson, 1966**) is chosen to decrease or suppress the tissue longitudinal magnetization relative to the thermal equilibrium value. The flowing blood water spins entering into the excited volume are fully relaxed, and as they flow through the imaging volume their longitudinal magnetization $M_z^{\text{blood}}$ just before the $n$th RF pulse is (**Brown et al., 2014a**):

$$M_z^{\text{blood}}\left(n_{\text{RF}}\right) = M_{zS}^{\text{blood}} + \left(\left(e^{-\frac{T_R}{T_1^{\text{blood}}}}\right) \cdot \cos\theta\right)^{n_{\text{RF}}-1} \cdot \left(M_0 - M_{zS}^{\text{blood}}\right). \tag{2}$$

The number of RF pulses experienced by the blood, $n_{\text{RF}} = t_{\text{delivery}}/T_R$, depends on the blood delivery time $t_{\text{delivery}}$, that is, the time it takes the blood spins from entering the imaging volume to reaching the target vessel, and the repetition time $T_R$. Hence, the longitudinal magnetization of the blood water spins is an exponentially decaying function of the number of applied RF pulses, and inversely related to the blood delivery time. We define the FRE, which is the fundamental contrast mechanism in TOF-MRA, as the difference in blood longitudinal magnetization and surrounding tissue signal relative to the tissue signal (**Al-Kwifi et al., 2002**):

$$\text{FRE}\left(n_{\text{RF}}\right) = \frac{M_z^{\text{blood}}\left(n_{\text{RF}}\right) - M_{zS}^{\text{tissue}}}{M_{zS}^{\text{tissue}}}. \tag{3}$$

In classical descriptions of the FRE effect (**Brown et al., 2014a**; **Carr and Carroll, 2012**), significant emphasis is placed on the effect of multiple 'velocity segments' within a slice in the 2D imaging case. Using the simplified plug-flow model, where the cross-sectional profile of blood velocity within the vessel is constant and effects such as drag along the vessel wall are not considered, these segments can be described as 'disks' of blood that do not completely traverse through the full slice within one $T_R$, and, thus, only a fraction of the blood in the slice is replaced. Consequently, estimation of the FRE effect would then need to accommodate contribution from multiple 'disks' that have experienced 1 to $k$ RF pulses. In the case of 3D imaging as employed here, multiple velocity segments within 1 voxel are generally not considered, as the voxel sizes in 3D are often smaller than the slice thickness in 2D imaging and it is assumed that the blood completely traverses through a voxel each $T_R$. However, the question arises whether this assumption holds for pial arteries, where blood velocity is

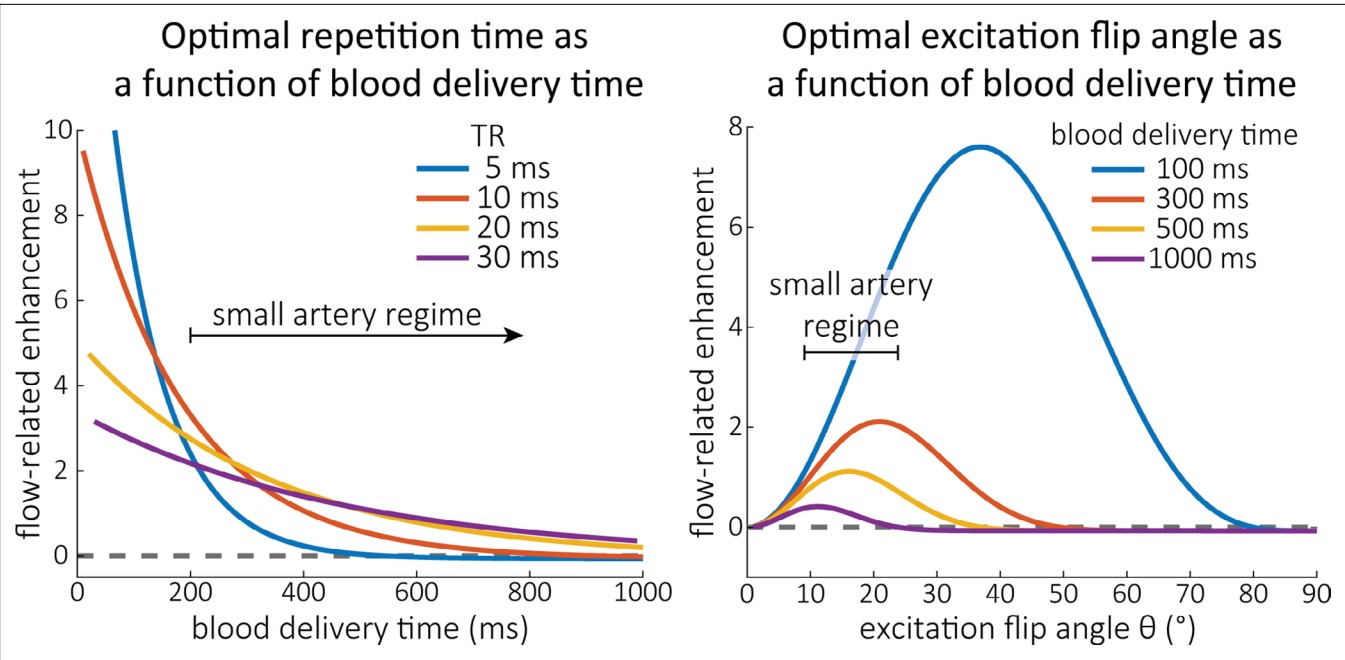

**Figure 3.** Optimal imaging parameters for small arteries. :Left: Flow-related enhancement (FRE) was simulated as a function of blood delivery time for different repetition times assuming an excitation flip angle of 18° and longitudinal relaxation times of blood and tissue of 2100 and 1950 ms at 7 Tesla (T), respectively. Overall, FRE decreases with increasing blood delivery times. In the small vessel regime, that is, at blood delivery times >200 ms, longer repetition times result in higher FRE than shorter repetition times. Right: FRE was simulated as a function of excitation flip angle for different blood delivery times assuming a $T_R$ value of 20 ms and longitudinal relaxation rates as above. The excitation flip angle that maximizes FRE for longer blood delivery times is lower than the excitation flip angle that maximizes FRE for shorter blood delivery times, and often the optimal excitation flip angle is close to the Ernst angle for longer blood delivery times (Ernst angle: 8.2°; optimal excitation flip angles: 37° (100 ms), 21° (300 ms), 16° (500 ms), 11° (1000 ms)).

The online version of this article includes the following figure supplement(s) for figure 3:

**Figure supplement 1.** Optimal imaging parameters for small arteries.

considerably lower than in intracranial arteries (*Figure 2*). To comprehensively answer this question, we have computed the *blood dwell time* (note that the blood dwell time is inversely related to the critical velocity in 2D MRA but does not need to assume a specific $T_R$ value), that is, the average time it takes the blood to traverse a voxel, as a function of blood velocity and voxel size (*Figure 2*). For reference, the blood velocity estimates from the three studies mentioned above *Bouvy et al., 2016*; *Kobari et al., 1984*; *Nagaoka and Yoshida, 2006* have been added in this plot as horizontal white lines. For the voxel sizes of interest here, that is, 50–300 μm, blood dwell times are, for all but the slowest flows, well below commonly used repetition times (*Brown et al., 2014a*; *Carr and Carroll, 2012*; *Ladd, 2007*; *Von Morze et al., 2007*). Thus, in a first approximation using the plug-flow model, it is not necessary to include several velocity segments for the voxel sizes of interest when considering pial arteries, as one might expect from classical treatments, and the FRE effect can be described by *Equations 1–3*, simplifying our characterization of FRE for these vessels. When considering the effect of more complex flow patterns, it is important to bear in mind that the arteries targeted here are only 1 voxel thick, and signals are integrated across the whole artery.

Based on these equations, optimal $T_R$ and excitation flip angle values ($\theta$) can be calculated for the blood delivery times under consideration (*Figure 3*). To better illustrate the regime of small arteries, we have illustrated the effect of either flip angle or $T_R$ while keeping the other parameter values fixed to the value that was ultimately used in the experiments; although both parameters can also be optimized simultaneously (*Haacke et al., 1990*). *Figure 3—figure supplement 1* further delineates the interdependency between flip angle and $T_R$ within a parameter range commonly used for TOF imaging at ultra-high field (*Kang et al., 2010*; *Stamm et al., 2013*; *Von Morze et al., 2007*). Note how longer $T_R$ values still provide an FRE effect even at very long blood delivery times, whereas using shorter $T_R$ values can suppress the FRE effect (*Figure 3*, left). Similarly, at lower flip angles, the FRE

effect is still present for long blood delivery times, but it is not available anymore at larger flip angles, which, however, would give maximum FRE for shorter blood delivery times (*Figure 3*, right). Due to the non-linear relationships of both blood delivery time and flip angle with FRE, the optimal imaging parameters deviate considerably when comparing blood delivery times of 100 and 300 ms, but the differences between 300 and 1000 ms are less pronounced. In the following simulations and measurements, we have thus used a $T_R$ value of 20 ms, that is, a value only slightly longer than the readout of the high-resolution TOF acquisitions, which allowed time-efficient data acquisition, and a nominal excitation flip angle of 18°. From a practical standpoint, these values are also favourable as the low flip angle reduces the SAR (*Fiedler et al., 2018*) and the long $T_R$ value decreases the potential for peripheral nerve stimulation (*Mansfield and Harvey, 1993*).

The optimizations presented so far have assumed that the voxel is fully filled with blood. However, in the case of pial arteries, vessel diameters are often comparable to or smaller than the voxel size. In the next section, we will therefore investigate the individual contributions of blood delivery times and voxel size to the overall FRE in pial arteries. We will introduce a model that consider the partial-volume effects of blood and tissue to allow us to decide whether imaging of pial arteries is limited by the physiological properties of the brain vasculature, that is, blood flow rates, or the available image resolution.

## Introducing a partial-volume model

To account for the effect of voxel volume on the FRE, the total longitudinal magnetization $M_z$ needs to also consider the number of spins contained within in a voxel (*Du et al., 1996*; *Venkatesan and Haacke, 1997*). A simple approximation can be obtained by scaling the longitudinal magnetization with the voxel volume (*Venkatesan and Haacke, 1997*). (Note that this does not affect the relative FRE definition presented in *Equation (3)*, as the effect of total voxel volume cancels out.) To then include partial-volume effects, the total longitudinal magnetization in a voxel $M_z^{\text{total}}$ becomes the sum of the contributions from the stationary tissue $M_{zS}^{\text{tissue}}$ and the inflowing blood $M_z^{\text{blood}}$, weighted by their respective volume fractions $V_{\text{rel}}$:

$$M_z^{\text{total}} = V_{\text{rel}}^{\text{blood}} \cdot M_z^{\text{blood}} + V_{\text{rel}}^{\text{tissue}} \cdot M_{zS}^{\text{tissue}}. \tag{4}$$

For simplicity, we assume a single vessel is located at the centre of the voxel and approximate it to be a cylinder with diameter $d_{\text{vessel}}$ and length $l_{\text{voxel}}$ of an assumed isotropic voxel along one side. The relative volume fraction of blood $V_{\text{rel}}^{\text{blood}}$ is the ratio of vessel volume within the voxel to total voxel volume (see section *Estimation of vessel-volume fraction* and *Figure 4—figure supplement 1*), and the tissue volume fraction $V_{\text{rel}}^{\text{tissue}}$ is the remainder that is not filled with blood, or

$$V_{\text{rel}}^{\text{tissue}} = 1 - V_{\text{rel}}^{\text{blood}}. \tag{5}$$

We can now replace the blood magnetization in *Equation (3)* with the total longitudinal magnetization of the voxel to compute the FRE as a function of vessel-volume fraction:

$$\text{FRE}_{\text{PV}}\left(n_{\text{RF}}, l_{\text{voxel}}, d_{\text{vessel}}\right) = \frac{M_z^{\text{total}}\left(n_{\text{RF}},\, l_{\text{voxel}},\, d_{\text{vessel}}\right) - M_{zS}^{\text{tissue}}}{M_{zS}^{\text{tissue}}}. \tag{6}$$

Note that this can also be written as $\text{FRE}_{\text{2C}}\left(n_{\text{RF}}, l_{\text{voxel}},\, d_{\text{vessel}}\right) = V_{\text{rel}}^{\text{blood}}\left(l_{\text{voxel}}, d_{\text{vessel}}\right) \cdot \text{FRE}\left(n_{\text{RF}}\right)$.

*Figure 4* illustrates the partial-volume FRE (FRE$_{\text{PV}}$) assuming a pial artery diameter ($d_{\text{vessel}}$) of 200 µm, a $T_R$ value of 20 ms, an excitation flip angle of 18°, and longitudinal relaxation times of blood and tissue of 2100 and 1950 ms, respectively (*Huber, 2014*, Table 2.1). Therein, two prominent regimes appear along the voxel size axis: (i) the blood-delivery-time dominated regime, where as long as the voxel is completely filled with blood, that is, $l_{\text{voxel}} < \cos\frac{\pi}{4} \cdot d_{\text{vessel}}$, the voxel size has no impact on the FRE, and the FRE is solely an exponential function of the blood delivery time (*Equation (2)*, *Figure 3*); (ii) the voxel-size dominated regime, where if the vessel is smaller than the voxel, the FRE also depends quadratically on the voxel size (see *Estimation of vessel-volume fraction*). Note that our partial-volume model encompasses both regimes, and because the FRE$_{\text{2C}}$ definition reverts to the classical FRE definition for larger-than-voxel-sized vessels, we will from now on use FRE synonymously with FRE$_{\text{2C}}$. In theory, a reduction in blood delivery time increases the FRE in both regimes, and—if the vessel is smaller than the voxel—so would a reduction in voxel size. In practice, a reduction in slab

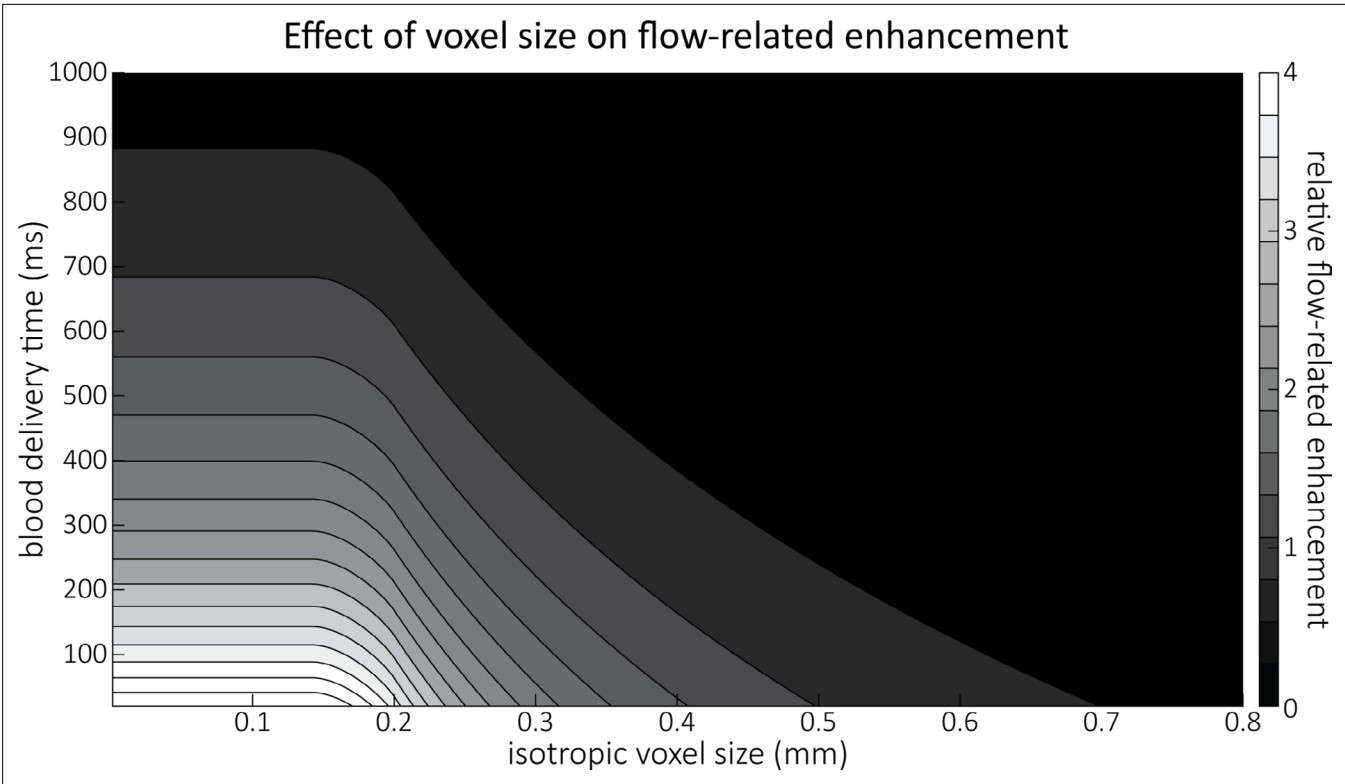

**Figure 4.** Effect of voxel size and blood delivery time on the relative flow-related enhancement (FRE), assuming a pial artery diameter of 200 µm. The relative FRE represents the signal intensity in the voxel containing the artery relative to the surrounding tissue (*Equation (3)*). For example, an FRE of 0 would indicate equal intensity, and an FRE of 1 would indicate twice the signal level in the voxel compared to the surrounding tissue. The FRE decreases with blood delivery time due to more signal attenuation (*Equation (2)*) and with voxel size due to more partial-volume effects (*Equation (4)*).

The online version of this article includes the following figure supplement(s) for figure 4:

**Figure supplement 1.** Estimation of the blood volume fraction for case (I) voxel in vessel, case (II) vessel intersects voxel, and case (III) vessel in voxel.

**Figure supplement 2.** Effect of voxel size and blood delivery time on the relative flow-related enhancement (FRE) using either a relative (**A,B**) (*Equation (3)*) or an absolute (**C,D**) (*Equation (12)*) FRE definition assuming a pial artery diameter of 200 µm (**A,C**) or 2000 µm, that is, no partial-volume effects at the central voxel of this artery considered here.

thickness—which is the default strategy in classical TOF-MRA to reduce blood delivery time—might not provide substantial FRE increases for pial arteries. This is due to their convoluted geometry (see section *Anatomical architecture of the pial arterial vasculature*), where a reduction in slab thickness may not necessarily reduce the vessel segment length if the majority of the artery is still contained within the smaller slab. Thus, given the small arterial diameter, reducing the voxel size is a much more promising avenue when imaging the pial arterial vasculature.

## Inflow artefacts in sinuses and pial veins

Inflow in large pial veins and the sagittal and transverse sinuses can cause FRE in these non-arterial vessels. One common strategy to remove this unwanted signal enhancement is to apply venous suppression pulses during the data acquisition, which saturate bloods spins outside the imaging slab. Disadvantages of this technique are the technical challenges of applying these pulses at ultra-high field due to constraints of the specific absorption rate (SAR) and the necessary increase in acquisition time (*Conolly et al., 1988*; *Heverhagen et al., 2008*; *Johst et al., 2012*; *Maderwald et al., 2008*; *Schmitter et al., 2012*; *Zhang et al., 2015*). In addition, optimal positioning of the saturation slab in the case of pial arteries requires further investigation, and in particular supressing signal from the superior sagittal sinus without interfering in the imaging of the pial arteries vasculature at the top of the cortex might prove challenging. Furthermore, this venous saturation strategy is based on the assumption that arterial blood is traveling head-wards while venous blood is drained foot-wards. For the complex and convoluted trajectory of pial vessels, this directionality-based saturation might be

oversimplified, particularly when considering the higher-order branches of the pial arteries and veins on the cortical surface.

Inspired by techniques to simultaneously acquire a TOF image for angiography and a susceptibility-weighted image (SWI) for venography (*Bae et al., 2010*; *Deistung et al., 2009*; *Du et al., 1994*; *Du and Jin, 2008*), we set out to explore the possibility of removing unwanted venous structures from the segmentation of the pial arterial vasculature during data post-processing. Because arteries filled with oxygenated blood have $T_2{}^*$ values similar to tissue, while veins have much shorter $T_2{}^*$ values due to the presence of deoxygenated blood (*Pauling and Coryell, 1936*; *Peters et al., 2007*; *Uludağ et al., 2009*; *Zhao et al., 2007*), we used this criterion to remove vessels with short $T_2{}^*$ values from the segmentation (see *Data analysis* for details). In addition, we also explored whether unwanted venous structures in the high-resolution TOF images—where a two-echo acquisition is not feasible due to the longer readout—can be removed based on detecting them in a lower-resolution image.

## Velocity- and $T_E$-dependent vessel displacement artefacts

While isotropic 3D TOF-MRA provides detailed information on the vessel position, the continuous motion of blood during image acquisition can lead to vessel displacement artefacts in the phase-encoding directions (*Brown et al., 2014b*; *Parker et al., 2003*). The magnitude of this misregistration artefact depends on the time elapsed between the time of the phase-encoding blip ($t_{pe}$) and the echo time ($T_E$) and on the blood velocity in the primary phase-encoding direction $y$, $v_y$:

$$\Delta y_{\mathrm{displ}} = -v_y \cdot \left( T_E - t_{pe} \right). \tag{7}$$

A similar relationship for $\Delta z_{\mathrm{displ}}$ exists for the secondary phase-encoding direction $z$. According to this relationship, and based on the reported velocity range in *Figure 2*, vessel displacements in the phase-encoding directions between 25 and 250 µm for a time delay ($T_E - t_{pe}$) of 5 ms can be expected. Flow compensation gradients can correct for the effect of constant velocities (*Parker et al., 2003*), but the required large gradient lobes for high-resolution applications would substantially increase the $T_E$, and, consequently, reduce the SNR. For these reasons, no flow compensation in the phase-encoding-directions was used for the high-resolution images in this study. However, we do assess the extent of vessel displacement using a two-echo low-resolution acquisition.

## Considerations for imaging the pial arterial vasculature

We have identified voxel size as the key parameter for imaging pial arteries. While the benefit of smaller voxels was conceptually known and empirically shown before (*Haacke et al., 1990*; *Mattern*

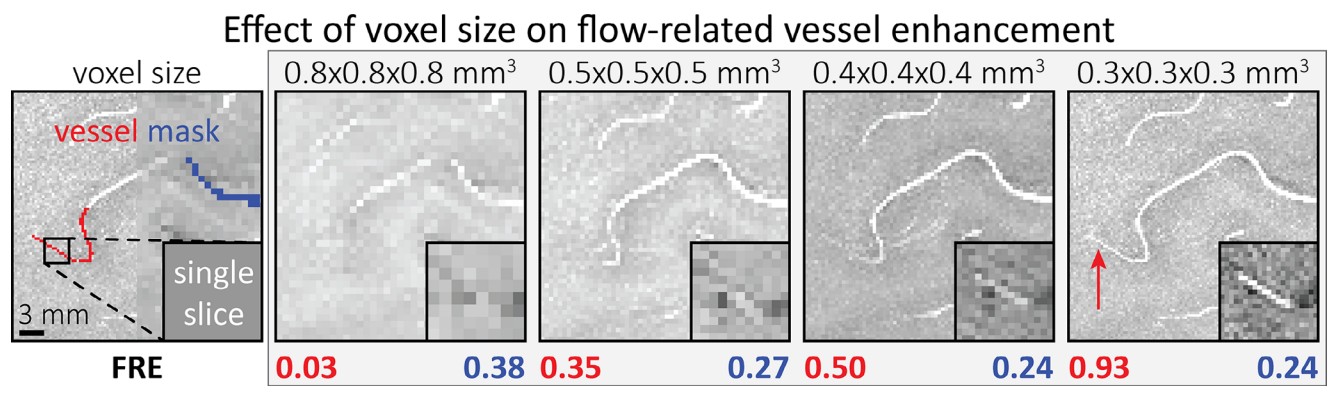

**Figure 5.** Effect of voxel size on flow-related vessel enhancement. Thin axial maximum intensity projections containing a small artery acquired with different voxel sizes ranging from 0.8 to 0.3 mm isotropic are shown. The flow-related enhancement (FRE) is estimated using the mean intensity value within the vessel masks depicted on the left, and the mean intensity values of the surrounding tissue. The small insert shows a section of the artery as it lies within a single slice. A reduction in voxel size is accompanied by a corresponding increase in FRE (red mask), whereas no further increase is obtained once the voxel size is equal or smaller than the vessel size (blue mask).

The online version of this article includes the following figure supplement(s) for figure 5:

**Figure supplement 1.** Increase of vessel skeleton length with voxel size reduction.

*et al., 2018*; *Von Morze et al., 2007*), clear guidelines on how small is sufficiently small and how strongly the FRE increases with decreasing voxel size were not established. We can now conclude that the voxel size should be at least the size of the pial artery, that is, somewhere between 50 and 280 μm in our case, and ideally around 70% of that, that is, 35–200 μm. While this is challenging in practice, the strong increase in FRE especially when voxel sizes are similar to vessel size makes a reduction of even only 10 or 20 μm worthwhile for imaging pial arteries. Further, we have seen that slow blood velocities are an unlikely explanation for loss of FRE in small vessels, because (i) blood delivery times to pial arteries are sufficiently short due to the perhaps surprisingly fast blood flow in these mesoscopic vessels, (ii) the effect of blood delivery time on FRE decreases with increasing blood delivery times, and (iii) the addition of partial-volume effects to the FRE estimation predicts the often observed drastic loss of contrast in small vessels. Thus, we expect a reduction in voxel size to directly translate into an increased sensitivity for small pial arteries.

## Results

### Effect of voxel size on FRE

To investigate the effect of voxel size on vessel FRE, we acquired data at four different voxel sizes ranging from 0.8 to 0.3 mm isotropic resolution, adjusting only the encoding matrix, with imaging parameters being otherwise identical (FOV, TR, TE, flip angle, *R*, slab thickness, see section *Data acquisition*). The total acquisition time increases from less than 2 min for the lowest resolution scan to over 6 min for the highest resolution scan as a result. *Figure 5* shows thin maximum intensity projections of a small artery. While the artery is not detectable at the largest voxel size, it slowly emerges as the voxel size decreases and approaches the vessel size. Presumably, this is driven by the considerable increase in FRE as seen in the single slice view (*Figure 5*, small inserts). Accordingly, the FRE computed from the vessel mask for the smallest part of the vessel (*Figure 5*, red mask) increases substantially with decreasing voxel size. More precisely, reducing the voxel size from 0.8, 0.5, or 0.4 mm to 0.3 mm increases the FRE by 2900%, 165%, and 85%, respectively. Assuming an arterial diameter of 300 μm partial-volume FRE model (*Introducing a partial-volume model*) would predict similar ratios of 611%, 178%, and 78%.

However, as long as the vessel is larger than the voxel (*Figure 5*, blue mask), the relative FRE does not change with resolution (see also *Effect of FRE definition and interaction with partial-volume model* and *Figure 4—figure supplement 2*). To illustrate the gain in sensitivity to detect smaller arteries, we have estimated the relative increase of the total length of the segmented vasculature (*Figure 5—figure supplement 1*): reducing the voxel size from 0.8 to 0.5 mm isotropic increases the skeleton length by 44%, reducing the voxel size from 0.5 to 0.4 mm isotropic increases the skeleton length by 28%, and reducing the voxel size from 0.4 to 0.3 mm isotropic increases the skeleton length by 31%. In summary, when imaging small pial arteries, these data support the hypothesis that it is primarily the voxel size, not blood delivery time, which determines whether vessels can be resolved.

### Imaging the pial arterial vasculature

Based on these results, we then proceeded to image the pial arterial vasculature at a much higher resolution of 0.16 mm isotropic, to verify that small pial arteries can be visualized using a TOF-based contrast. *Figure 6* shows the axial maximum intensity projection overlaid with a vessel segmentation on the left, a coronal projection of the central part of the brain, a sagittal projection and a 3D view of the segmentation on the right. Note the numerous right-angled branches emerging from the parent artery, indicating the increased sensitivity to small pial arteries, and the overall high vessel density compared with acquisitions performed at conventional resolutions (*Bernier et al., 2018*; *Brown et al., 2014a*; *Chen et al., 2018*; *Ladd, 2007*; *Mattern et al., 2018*; *Schulz et al., 2016*; *Stucht et al., 2015*; *Von Morze et al., 2007*; *Wright et al., 2013*).

The horizontal lines in the background presumably stem from stacking multiple slabs containing slightly different imaging volumes and potentially imperfect slab profiles. Data acquired with the same parameters in an additional participant is shown in *Figure 6—figure supplement 1*. For reference, data from both participants without the overlay of the segmentation are presented in *Figure 6—figure supplements 2 and 3*.

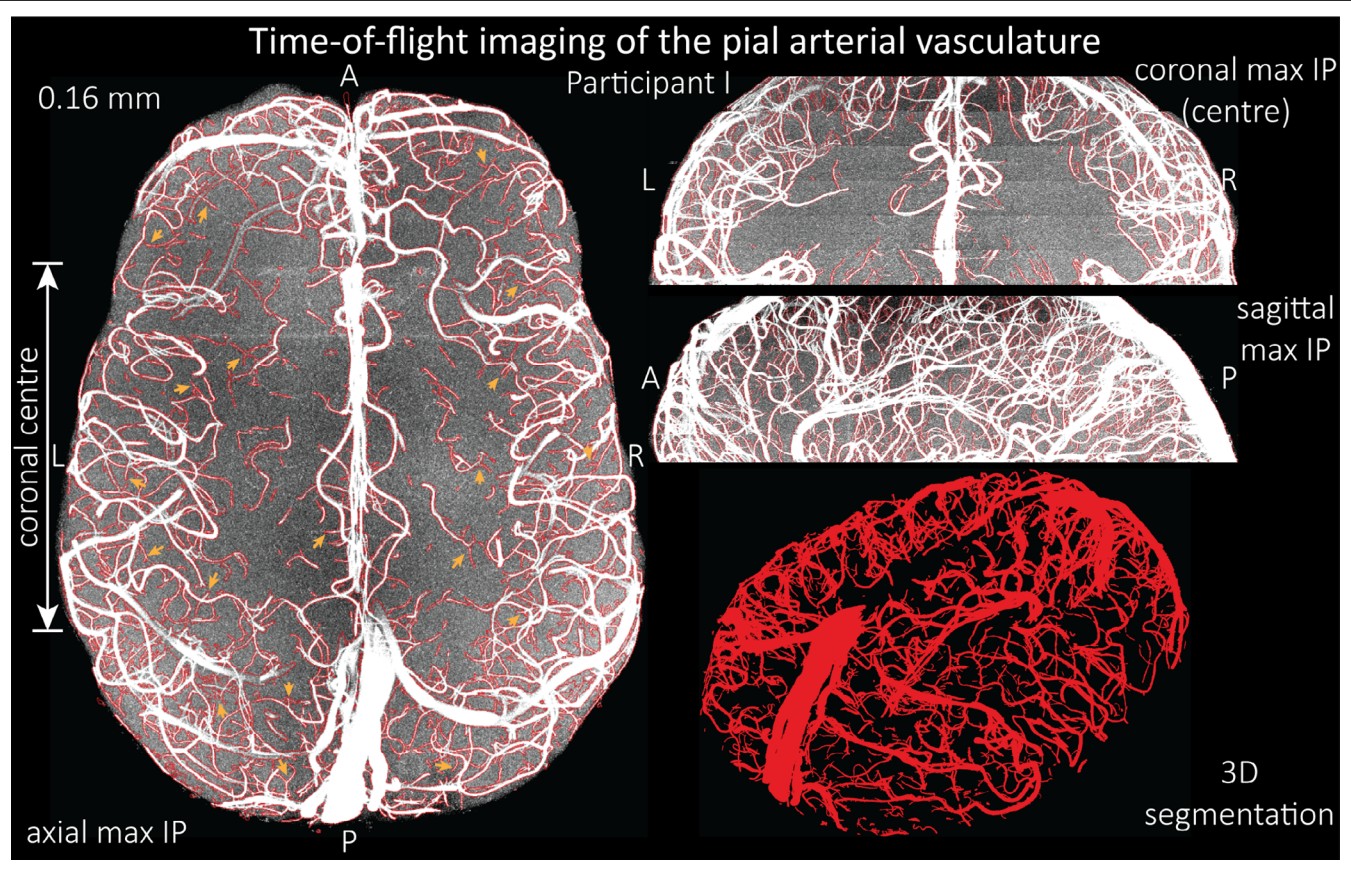

**Figure 6.** Time-of-flight imaging of the pial arterial vasculature at 0.16 mm isotropic resolution and 50 mm coverage in the head-foot direction. Left: Axial maximum intensity projection and outline of the vessel segmentation overlaid in red. Examples of the numerous right-angled branches are indicated by orange arrows. Right: Coronal maximum intensity projection and segmentation of the central part of the brain (top), sagittal maximum intensity projection and segmentation (middle), and 3D view of the vessel segmentation (bottom).

The online version of this article includes the following video and figure supplement(s) for figure 6:

**Figure supplement 1.** Time-of-flight imaging of the pial arterial vasculature at 0.16 mm isotropic resolution and 58 mm coverage in the head-foot direction showing the results from participant III using the same acquisition protocol as for *Figure 6*.

**Figure supplement 2.** Time-of-flight imaging of the pial arterial vasculature at 0.16 mm isotropic resolution and 50 mm coverage in the head-foot direction showing the same data of participant I as in *Figure 6*, but without the overlay of the segmentation.

**Figure supplement 3.** Time-of-flight imaging of the pial arterial vasculature at 0.16 mm isotropic resolution and 58 mm coverage in the head-foot direction showing the same data of participant III as in *Figure 6—figure supplement 1* but without the overlay of the segmentation.

**Figure 6—video 1.** Video of the segmentation of the time-of-flight data acquired with 160 μm isotropic voxel size as shown in Figure 6.
https://elifesciences.org/articles/71186/figures#fig6video1

**Figure 6—video 2.** Video of the segmentation of the time-of-flight data acquired with 160 μm isotropic voxel size as shown in Figure 6—figure supplement 1.
https://elifesciences.org/articles/71186/figures#fig6video2

To assess the benefit of even higher resolution, we acquired TOF data with an isotropic voxel size of 140 μm in a second participant using prospective motion correction (*Mattern et al., 2018*) to reduce motion artefacts given the even longer acquisition time and higher resolution (*Figure 7*). Again for reference, data without the overlay of the segmentation are presented in *Figure 7—figure supplement 1*.

The axial maximum intensity projection illustrates the large number of right-angled branches visible at this resolution. Similarly, the coronal and sagittal projections demonstrate the high density of the vasculature at this level. Note that unwanted signal in non-arterial vessels such as pial veins and the superior sagittal sinus has been removed using an auxiliary acquisition as described in the section *Removal of pial veins* below. We have also acquired one single slab with an isotropic voxel size of

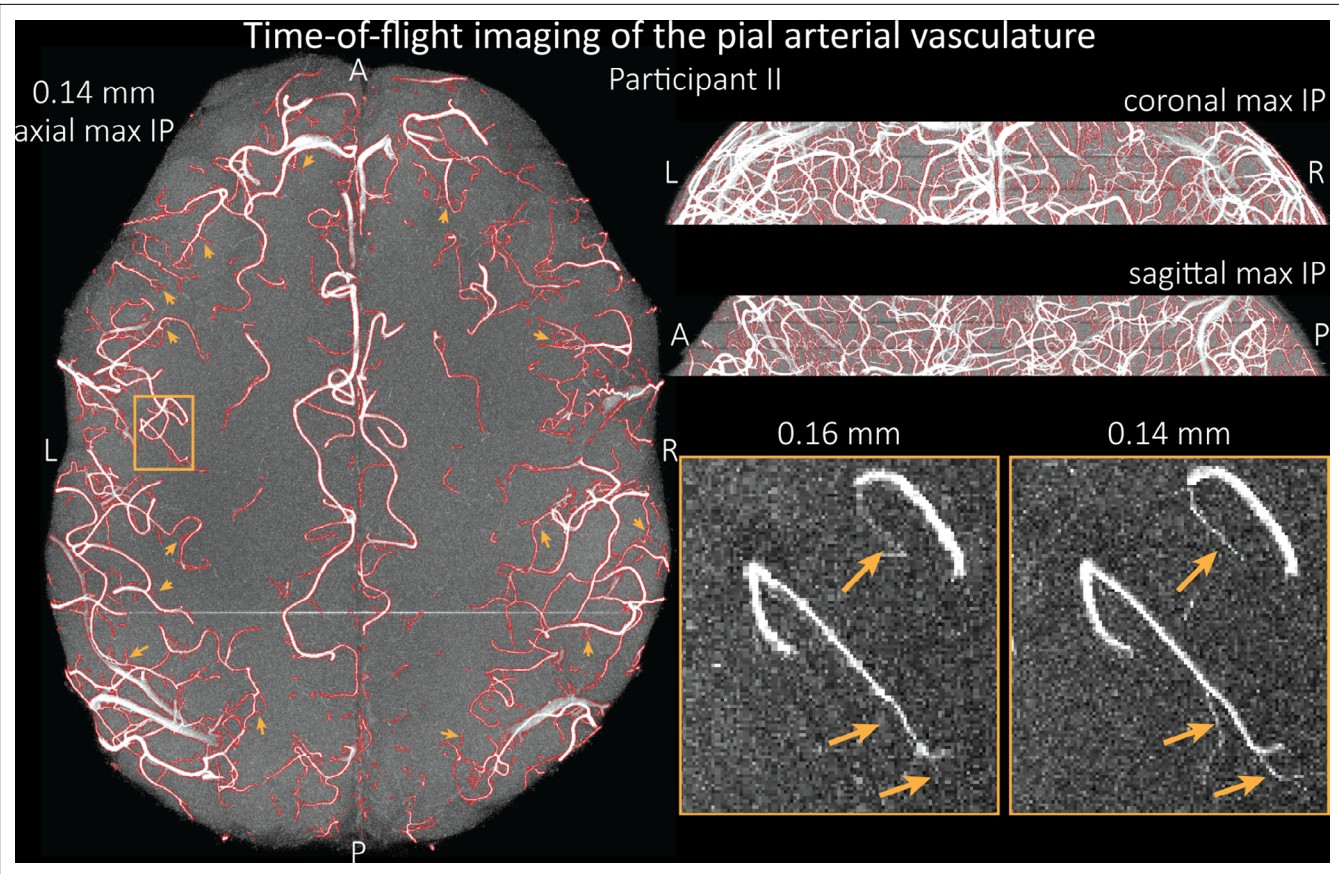

**Figure 7.** Time-of-flight imaging of the pial arterial vasculature at 0.14 mm isotropic resolution and 19.6 mm coverage in the foot-head direction. Left: Axial maximum intensity projection and outline of the vessel segmentation overlaid in red. Examples of the numerous right-angled branches are indicated by orange arrows. Right: Coronal maximum intensity projection and segmentation (top) and sagittal maximum intensity projection and segmentation (middle). A comparison of 0.16 and 0.14 mm voxel size (bottom) shows several vessels visible only at the higher resolution. The location of the insert is indicated on the axial maximum intensity projection on the left. Note that for this comparison, the coverage of the 0.14 mm data was matched to the smaller coverage of the 0.16 mm data.

The online version of this article includes the following video and figure supplement(s) for figure 7:

**Figure supplement 1.** Time-of-flight imaging of the pial arterial vasculature at 0.14 mm isotropic resolution and 19.6 mm coverage in the foot-head direction showing the same data of participant II as in *Figure 7*, but without the overlay of the segmentation.

**Figure supplement 2.** Additional examples comparing the flow-related enhancement at 0.14 mm (top) and 0.16 mm (bottom) voxel size showing additional small branches visible only at the higher resolution.

**Figure supplement 3.** Small section of the time-of-flight data (13.44 mm × 9.94 mm × 7 mm) acquired at 0.14 mm isotropic resolution and the corresponding segmentation of the pial arterial vasculature from *Figure 7* showing the performance of the automatic segmentation used in this study (left) and manual segmentation (right).

**Figure 7—video 1.** Video of the segmentation of the time-of-flight data acquired with 140 μm isotropic voxel size as shown in Figure 7. https://elifesciences.org/articles/71186/figures#fig7video1

0.16 mm with prospective motion correction for this participant in the same session to compare to the acquisition with 0.14 mm isotropic voxel size and to test whether any gains in FRE are still possible at this level of the vascular tree. Indeed, the reduction in voxel volume by 33% revealed additional small branches connected to larger arteries (see also *Figure 7—figure supplement 2*). For this example, we found an overall increase in skeleton length of 14% (see also *Figure 5—figure supplement 1*). In summary, we have found that reducing voxel sizes increases FRE up to at least 0.14 mm isotropic resolution and conclude that partial-volume effects are the main contributor to FRE loss in small arteries.

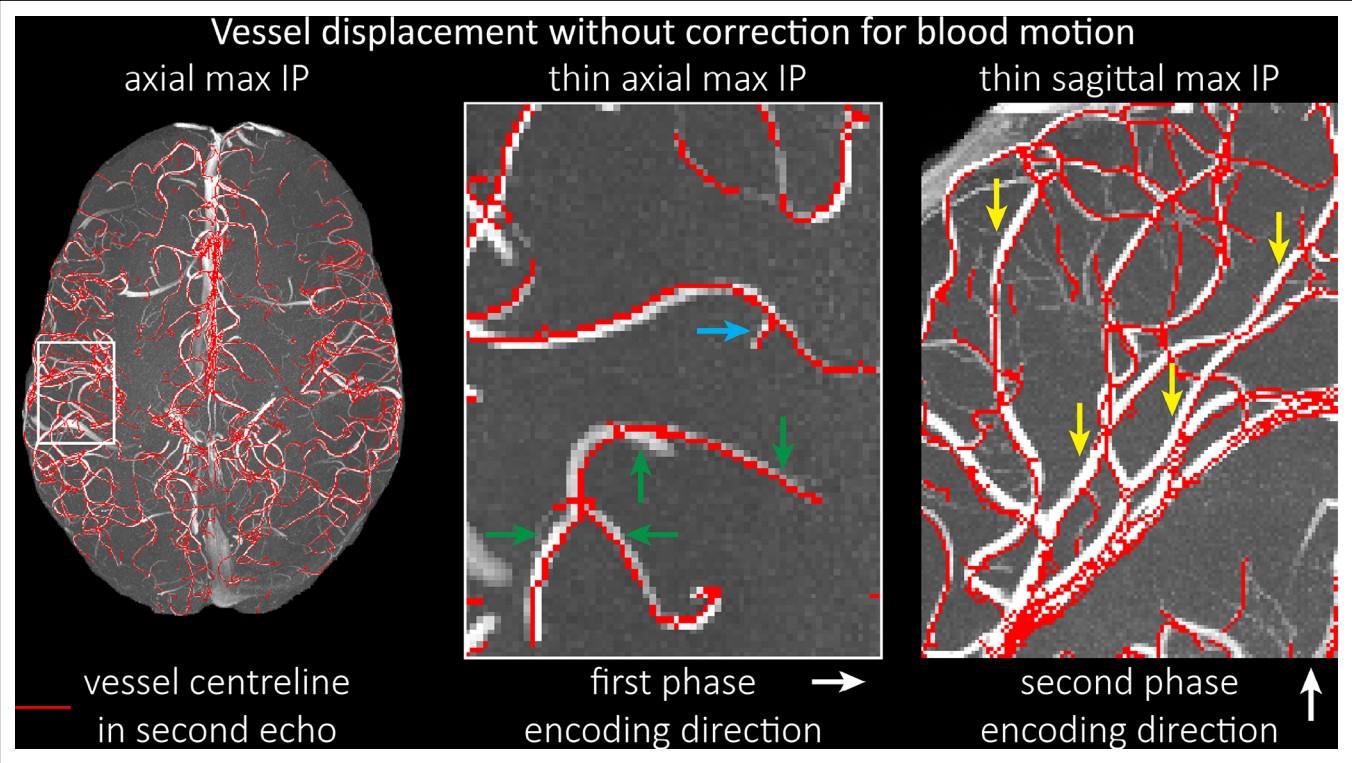

**Figure 8.** Vessel displacement without correction for blood motion. The vessel displacement is illustrated using a maximum intensity projection of the first, flow compensated echo of a two-echo time-of-flight (TOF) acquisition with the vessel centreline of the second echo without flow compensation overlaid in red. Strong displacements in both phase-encoding directions are present resulting in complex vessel shift patterns (green arrows). While furthest vessel displacements are observed in large arteries with faster flow (yellow arrows), considerable displacements arise even in smaller arteries (blue arrow).

The online version of this article includes the following video and figure supplement(s) for figure 8:

**Figure supplement 1.** Explanation of the vessel displacement artefact.

**Figure 8—video 1.** Illustration of the vessel displacement artefact showing axial maximum intensity projections of the first and second echo of the two-echo time-of-flight (TOF) in Figure 8.

https://elifesciences.org/articles/71186/figures#fig8video1

**Figure 8—video 2.** Illustration of the vessel displacement artefact showing sagittal maximum intensity projections of the first and second echo of the two-echo time-of-flight (TOF) in Figure 8.

https://elifesciences.org/articles/71186/figures#fig8video2

## Vessel displacement artefacts

The vessel displacement artefacts in the phase-encoding directions due to blood motion are illustrated in *Figure 8* using a two-echo TOF acquisition from a fourth participant. At a delay time of 10 ms between phase encoding and echo time, the observed displacement of approximately 2 mm in some of the larger arteries would correspond to a blood velocity of 200 mm/s, which is well within the expected range (*Figure 2*). For the smallest arteries, a displacement of 1 voxel (0.4 mm) can be observed, indicative of blood velocities of 40 mm/s. Note that the vessel displacement can be observed in all arteries visible at this resolution, indicating high blood velocities throughout much of the pial arterial vasculature. Thus, assuming a blood velocity of 40 mm/s (*Figure 2*) and a delay time of 5 ms for the high-resolution acquisitions (*Figure 6*), vessel displacements of 0.2 mm are possible, representing a shift of 1–2 voxels.

## Removal of pial veins

Inflow in large pial veins and the superior sagittal and transverse sinuses can cause an FRE in these non-arterial vessels (*Figure 9*, left). The higher concentration of deoxygenated haemoglobin in these vessels leads to shorter $T_2{}^*$ values (*Pauling and Coryell, 1936*), which can be estimated using a

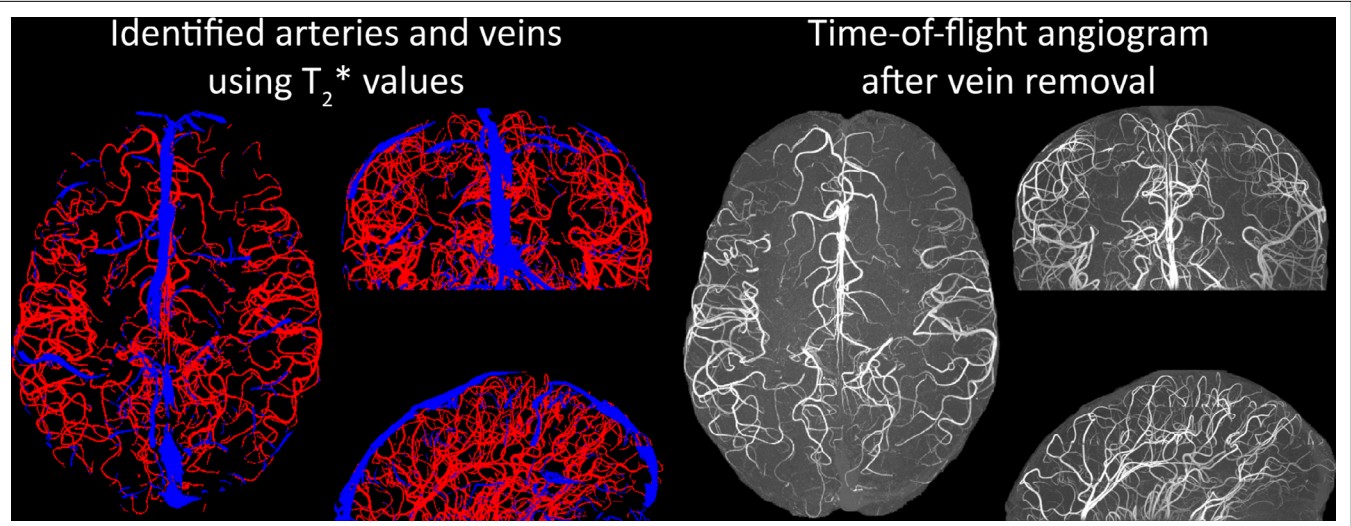

**Figure 9.** Removal of non-arterial vessels in time-of-flight imaging. Left: Segmentation of arteries (red) and veins (blue) using $T_2^*$ estimates. Right: Time-of-flight angiogram after vein removal.

two-echo TOF acquisition (see also *Inflow artefacts in sinuses and pial veins*). These vessels can be identified in the segmentation based on their $T_2$* values (*Figure 9*, left), and removed from the angiogram (*Figure 9*, right) (*Bae et al., 2010*; *Deistung et al., 2009*; *Du et al., 1994*; *Du and Jin, 2008*). Predominantly, large vessels which exhibited an inhomogeneous intensity profile and a steep loss of intensity at the slab boundary were identified as non-arterial (*Figure 9*, left). Further, we also explored the option of removing unwanted venous vessels from the high-resolution TOF image (*Figure 7*) using a low-resolution two-echo TOF (not shown). This indeed allowed us to remove the strong signal enhancement in the sagittal sinuses and numerous larger veins, although some small veins, which are characterized by inhomogeneous intensity profiles and can be detected visually by experienced raters, remain.

## Discussion

### A new perspective on imaging the pial arterial vasculature

We have outlined the theoretical components and provided empirical evidence for geometrically accurate imaging of the pial arterial vasculature of the human brain in vivo. We found that reducing the voxel size increases the vessel contrast when imaging the pial arterial vasculature of the human brain, in contrast to previous theoretical and empirical treatments of the effect of voxel size on vessel contrast (*Du et al., 1993*; *Du et al., 1996*; *Venkatesan and Haacke, 1997*). Further, we could not confirm the common assumption that slow blood flow is the main limiting factor for high-resolution TOF imaging (*Chen et al., 2018*; *Haacke et al., 1990*; *Masaryk et al., 1989*; *Mut et al., 2014*; *Park et al., 2020*; *Parker et al., 1991*; *Wilms et al., 2001*; *Wright et al., 2013*), but instead found adequate signal levels even in smallest pial arteries given sufficiently small voxels (*Figure 7*). Both effects are driven by non-linear relationships which might have contributed to their over- and under-estimation in the literature. In short, FRE depends exponentially on blood delivery times (*Figure 3*), such that it *decreases* for longer blood delivery times, and quadratically on voxel size (*Figure 4*), such that it *increases* for smaller voxels—that is, the imaging regime of pial arteries.

### Extending classical FRE treatments to the pial vasculature

There are several major modifications in our approach to this topic that might explain why, in contrast to predictions from classical FRE treatments, it is indeed possible to image pial arteries. For instance, the definition of vessel contrast or FRE is often stated as an absolute difference between blood and tissue signal (*Brown et al., 2014a*; *Carr and Carroll, 2012*; *Du et al., 1993*; *Du et al., 1996*; *Haacke et al., 1990*; *Venkatesan and Haacke, 1997*). Here, however, we follow the approach of *Al-Kwifi*

*et al., 2002*, and consider *relative* contrast. While this distinction may seem to be semantic, the effect of voxel volume on FRE for these two definitions is exactly opposite: *Du et al., 1996*, concluded that larger voxel size increases the (absolute) vessel-background contrast, whereas here we predict an increase in relative FRE for small arteries with decreasing voxel size. Therefore, predictions of the depiction of small arteries with decreasing voxel size differ depending on whether one is considering absolute contrast, that is, difference in longitudinal magnetization, or relative contrast, that is, contrast differences independent of total voxel size. Importantly, this prediction changes for large arteries where the voxel contains only vessel lumen, in which case the relative FRE remains constant across voxel sizes, but the absolute FRE increases with voxel size (*Figure 4—figure supplement 2*). Overall, the interpretations of relative and absolute FRE differ, and one measure may be more appropriate for certain applications than the other. Absolute FRE describes the difference in magnetization and is thus tightly linked to the underlying physical mechanism. Relative FRE, however, describes the image contrast and segmentability. If blood and tissue magnetization are equal, both contrast measures would equal zero and indicate that no contrast difference is present. However, when there is signal in the vessel and as the tissue magnetization approaches zero, the absolute FRE approaches the blood magnetization (assuming no partial-volume effects), whereas the relative FRE approaches infinity. While this infinite relative FRE does not directly relate to the underlying physical process of 'infinite' signal enhancement through inflowing blood, it instead characterizes the segmentability of the image in that an image with zero intensity in the background and non-zero values in the structures of interest can be segmented perfectly and trivially. Accordingly, numerous empirical observations (*Al-Kwifi et al., 2002*; *Bouvy et al., 2014*; *Haacke et al., 1990*; *Ladd, 2007*; *Mattern et al., 2018*; *Von Morze et al., 2007*) including the data provided here (*Figures 5–7*) have shown the benefit of smaller voxel sizes if the aim is to visualize and segment small arteries.

In addition, we also formulated the problem in 3D space taking into account the spatial characteristics of blood vessels in particular, which are in principle elongated, and approximately 1D structures. Because *Venkatesan and Haacke, 1997*, assumed a 1D image consisting of a row of samples, they predicted *no* impact of voxel size on absolute FRE, despite using the same theory as *Du et al., 1996*, who operated in 2D space and predicted increased FRE for larger voxel sizes. The image intensity of a voxel with an elongated vessel in its centre will scale differently with increasing isotropic voxel size than a voxel containing point-like (0D) or extended sheet-like (2D) objects. Note that classical considerations of partial-volume effects (*Shattuck et al., 2001*) cannot be readily applied to vessels, as these treatments usually assume 2D borders of larger structures meeting in a voxel (such as the boundary between two tissue volumes). Thus, not only the FRE definition in terms of absolute or relative contrast, but also the dimensionality and geometry of the assumed tissue content matters. In summary, we have presented a comprehensive theoretical framework for imaging small pial arteries which considers contrast and dimensionality, and from which we can conclude that small voxel sizes are the key ingredient for imaging the pial arterial vasculature.

## Imaging limitations

To maximize the sensitivity and accuracy of imaging small pial arteries, challenges regarding vessel localization and imaging noise remain to be investigated. First, the effect of vessel location within the voxel on the FRE needs to be addressed in more detail. Note that in our theoretical treatment we have always assumed a vessel located centrally in the voxel. If a vessel is not centred and instead was split over several voxels, such that its cross-section intersected the voxel border, we would, due to the reduction in relative blood volume fraction (*Equation (4)*), expect a reduction in FRE in each containing voxel. However, given that in Fourier imaging used in MRI the voxel grid is arbitrarily placed, previous work (*Du et al., 1994*; *Zhu et al., 2013*) has argued that zero-filling could resolve this issue, and would essentially 're-centre' the vessel within a voxel. However, it remains unclear what the optimal zero-filling factor would be. Classical Fourier theory predicts sensitivity gains only up to a zero-filling factor of 2; beyond that, zero-filling is equivalent to interpolation and correlation between samples is introduced (*Bartholdi and Ernst, 1973*). Nevertheless, slight qualitative improvements in image appearance have been reported for higher zero-filling factors (*Du et al., 1994*), presumably owing to a smoother representation of the vessels (*Bartholdi and Ernst, 1973*). In contrast, *Mattern et al., 2018*, reported no improvement in vessel contrast for their high-resolution data. Ultimately, for each application, for example, visual evaluation vs. automatic segmentation, the optimal zero-filling

factor needs to be determined, balancing image appearance (*Du et al., 1994*; *Zhu et al., 2013*) with loss in statistical independence of the image noise across voxels. For example, in *Figure 5*, when comparing across different voxel sizes, the visual impression might improve with zero-filling. However, it remains unclear whether the same zero-filling factor should be applied for each voxel size, which means that the overall difference in resolution remains, namely a nearly 20-fold reduction in voxel volume when moving from 0.8 mm isotropic to 0.3 mm isotropic voxel size. Alternatively, the same 'zero-filled' voxel sizes could be used for evaluation, although then nearly 94% of the samples used to reconstruct the image with 0.8 mm voxel size would be zero-valued for a 0.3 mm isotropic resolution. Consequently, all data presented in this study were reconstructed without zero-filling.

A further challenge for obtaining accurate vessel localization is the vessel displacement (*Figure 8*) in both phase-encoding directions due to blood motion (*Figure 4*), which restricts the ability to register these images of the arterial vasculature with data from other image modalities. While modifications to the pulse sequence can address this problem (*Parker et al., 2003*), their practical implementation (e.g. peripheral nerve stimulation; *Mansfield and Harvey, 1993*) is limited in the case of small pial arteries (e.g. limits imposed by maximum gradient slew rates or peripheral nerve stimulation; *Mansfield and Harvey, 1993*), because the high-resolution acquisitions would require strong flow compensation gradients to be played out in a very short amount of time to ensure sufficiently short echo times needed to retain SNR. Classical approaches to correct apparent tissue displacement stemming from $B_0$-inhomogeneities such as reversal of the phase-encoding directions (*Andersson et al., 2003*) cannot be applied here, as the vessel displacement reflects true object motion (*Figure 8— figure supplement 1*). Note that correction techniques exist to remove displaced vessels from the image (*Gulban et al., 2021*), but they cannot revert the vessels to their original location. Alternatively, this artefact could also potentially be utilized as a rough measure of blood velocity. Other encoding strategies, such as radial and spiral acquisitions, experience no vessel displacement artefact because phase and frequency encoding take place in the same instant, although a slight blur might be observed instead (*Nishimura et al., 1995*; *Nishimura et al., 1991*). However, both trajectories pose engineering challenges and much higher demands on hardware and reconstruction algorithms than the Cartesian readouts employed here (*Kasper et al., 2018*; *Shu et al., 2016*), particularly to achieve 3D acquisitions with 160 μm isotropic resolution. Provided that the imaging echo times are kept short, the overall size of this effect is small for pial arteries and estimated to be less than 2 voxel lengths for the imaging protocols used in this study. However, the displacement of feeding arteries with faster velocities can reach values of 1 mm or more (*Figure 7*), introducing a non-linear distortion of the arteries depending on the blood velocity and direction. This may be inconsequential for many applications such as those investigating topological features, but may prove challenging when, for example, the aim is to accurately combine the segmentation of the pial vasculature with a segmentation of the cerebral cortical surface.

From a practical perspective when performing the imaging experiments, the required long acquisition times and small voxels make these high-resolution acquisitions particularly vulnerable to image artefacts from subject motion. This has been addressed utilizing a prospective motion correction system (*Mattern et al., 2018*) to enable the acquisition of TOF data with 0.14 mm isotropic voxel size and over 20 min acquisition time per imaging slab. This allowed for the successful correction of head motion of approximately 1 mm over the 60 min scan session, showing the potential of prospective motion correction at these very high resolutions. Note that for the comparison in *Figure 7*, one slab with 0.16 mm voxel size was acquired in the same session also using the prospective motion correction system. However, for the data shown in *Figure 6* and *Figure 6—figure supplement 1*, no prospective motion correction was used, and we instead relied on the experienced participants who contributed to this study. We found that the acquisition of TOF data with 0.16 mm isotropic voxel size in under 12 min acquisition time per slab possible without discernible motion artefacts, although even with this nearly ideal subject behaviour approximately 1 in 4 scans still had to be discarded and repeated. Motion correction will generally be necessary to acquire data at higher resolution than that presented here. Alternatively, imaging acceleration techniques could be applied (*Candes et al., 2006*; *Griswold et al., 2002*; *Pruessmann et al., 1999*), but because these techniques undersample the imaging data, they come at the cost of lower image SNR and significant reductions in scan times might not be possible. Acquiring these data at even higher field strengths would boost SNR (*Edelstein et al., 1986*; *Pohmann et al., 2016*) to partially compensate for SNR losses due to acceleration

and may enable faster imaging and/or smaller voxel sizes. This could facilitate the identification of the ultimate limit of the FRE effect and identify at which stage of the vascular tree does the blood delivery time become the limiting factor. While *Figure 7* indicates the potential for voxel sizes below 0.16 mm, the singular nature of this comparison warrants further investigations.

In general, we have not considered SNR, but only FRE, that is, the (relative) image contrast, assuming that segmentation algorithms would benefit from higher contrast for smaller arteries. Importantly, the acquisition parameters available to maximize FRE are limited, namely repetition time, flip angle, and voxel size. SNR, however, can be improved via numerous avenues independent of these parameters (*Brown et al., 2014c*; *Du et al., 1996*; *Heverhagen et al., 2008*; *Parker et al., 1991*; *Triantafyllou et al., 2011*; *Venkatesan and Haacke, 1997*), the simplest being longer acquisition times. If the aim is to optimize a segmentation outcome for a given acquisition time, the trade-off between contrast and SNR for the specific segmentation algorithm needs to be determined (*Klepaczko et al., 2016*; *Lesage et al., 2009*; *Moccia et al., 2018*; *Phellan and Forkert, 2017*). Our own—albeit limited—experience has shown that segmentation algorithms (including manual segmentation) can accommodate a perhaps surprising amount of noise using prior knowledge and neighbourhood information, making these high-resolution acquisitions possible. Importantly, note that our treatment of the FRE does not suggest that an arbitrarily small voxel size is needed, but instead that voxel sizes appropriate for the arterial diameter of interest are beneficial (in line with the classic 'matched-filter' rationale; *North, 1963*). Voxels smaller than the arterial diameter would not yield substantial benefits (*Figure 5*) and may result in SNR reductions that would hinder segmentation performance. Nevertheless, if SNR were also to be considered in the regime of pial arteries, the noise 'magnitude bias' due to the effects of multi-channel magnitude-valued data on the noise distribution (*Constantinides et al., 1997*; *Triantafyllou et al., 2011*) needs to be accounted for. In particular, the low SNR expected in the tissue signal, which in TOF-MRA should ideally be near the noise floor especially for high-resolution acquisitions, would be affected. Accordingly, SNR predictions then need to include the effects of various analog and digital filters, the number of acquired samples, the noise covariance correction factor, and—most importantly—the non-central chi distribution of the noise statistics of the final magnitude image (*Triantafyllou et al., 2011*).

Another imaging parameter that affects the FRE but has not been further considered here is the slab thickness. Classically, a reduced slab thickness is associated with higher FRE due to a reduction in blood delivery time (*Parker et al., 1991*). However, this assumes that the vessel runs relatively straight through the imaging volume. This will not always hold true when imaging pial arteries, which have numerous right-angled branches and track the folding pattern of the cortex. In addition, blood velocities in larger arteries are much faster, and most of the blood delivery time can be assumed to be spent in the small branches, which would be included in both thicker and thinner slabs. Nevertheless, we have used comparatively thin slabs due to the long acquisition times and higher risk of motion that would be associated with larger imaging volumes. Future acquisitions might be able to utilize the SNR increase from larger imaging volumes to increase acceleration and thus provide larger coverage at similar acquisition times.

In summary, numerous theoretical and practical considerations remain for optimal imaging of pial arteries using TOF contrast. Depending on the application, advanced displacement artefact compensation strategies may be required, and zero-filling could provide better vessel depiction. Further, an optimal trade-off between SNR, voxel size, and acquisition time needs to be found. Currently, the partial-volume FRE model only considers voxel size, and—as we reduced the voxel size in the experiments—we (partially) compensated the reduction in SNR through longer scan times. This, ultimately, also required the use of prospective motion correction to enable the very long acquisition times necessary for 140 µm isotropic voxel size. Often, anisotropic voxels are used to reduce acquisition time and increase SNR while maintaining in-plane resolution. This may indeed prove advantageous when the (also highly anisotropic) arteries align with the anisotropic acquisition, for example, when imaging the large supplying arteries oriented mostly in the head-foot direction. In the case of pial arteries, however, there is not preferred orientation because of the convoluted nature of the pial arterial vasculature encapsulating the complex folding of the cortex (see section *Anatomical architecture of the pial arterial vasculature*). A further reduction in voxel size may be possible in dedicated research settings utilizing even longer acquisition times and/or larger acquisition volumes to maintain SNR.

However, if acquisition time is limited, voxel size and SNR need to be carefully balanced against each other.

## Challenges for vessel segmentation algorithms

The vessel segmentations presented here were performed to illustrate the sensitivity of the image acquisition to small pial arteries and are based on a simple combination of thresholding and region-growing. Thus, there is much potential for increased detection sensitivity and accuracy by employing more sophisticated approaches (*Bernier et al., 2018*; *Chen et al., 2018*; *Frangi et al., 1998*; *Hsu et al., 2019*; *Hsu et al., 2017*; *Lesage et al., 2009*; *Nowinski et al., 2011*; *Suri et al., 2002*). *Figure 7—figure supplement 3* provides an example of this potential using manual segmentation on a small patch of the data presented in *Figure 7*. Given that the *manual* segmentation of these vessels is relatively simple, albeit arduous, machine learning approaches (*Hilbert et al., 2020*; *Tetteh et al., 2020*) also seem promising, as they commonly perform well in visual tasks (*LeCun et al., 2015*; *Rueckert et al., 2016*; *Zaharchuk et al., 2018*) and have successfully been applied to large-scale vasculature segmentations of mouse tissue-cleared data (*Todorov et al., 2020*). In general, the main challenges for these algorithms include the small vessel size compared to the voxel size, the broad range of vessel diameters, and the high noise levels.

Further, inflow in large pial veins and the dural venous sinuses can lead to high image intensities in these venous structures, and, consequently, false positives in the segmentation and identification of pial arteries. While additional RF pulses (*Meixner et al., 2019*; *Schmitter et al., 2012*) can be played out during image acquisition to suppress venous signal, the associated higher power deposition and increased acquisition times would reduce the imaging efficiency. Thus, we instead explored the removal of unwanted veins using a low-resolution two-echo TOF acquisition and $T_2^*$ estimates to identify non-arterial vessels in the segmentation of the high-resolution data (*Bae et al., 2010*; *Deistung et al., 2009*; *Du et al., 1994*; *Du and Jin, 2008*).

The success of this vein removal approach hinges on two conditions: the quality of the segmentation, that is, each segmented vessel is either an artery or a vein and no erroneous connections between the two classes arose during segmentation, and the assumption that venous vessels are present in both low- and high-resolution acquisitions. While the removal based on segmenting low-resolution data proved to be sufficient to demonstrate this approach (*Figure 9*), the segmentation of the high-resolution data suffered from a number of instances where veins and arteries were artefactually joined, and prohibited the full application of this technique to the native high-resolution data. Note the posterior lateral veins on both sides of the brain present in *Figure 7*, which require a higher exclusion threshold (see section *Data analysis*) to prevent the removal of joined arteries and veins. Thus, utilizing the numerous possible improvements in the segmentation algorithm (see above), these segmentation errors can be prevented in the future to improve vein removal for high-resolution data.

Our approach also assumes that the unwanted veins are large enough that they are also resolved in the low-resolution image. If we consider the source of the FRE effect and the type of vessels identified in *Figure 9*, it might indeed be exclusively large veins that are present in TOF-MRA data, which would suggest that our assumption is valid. Fundamentally, the FRE depends on the inflow of unsaturated spins into the imaging slab. However, small veins drain capillary beds in the local tissue, that is, the tissue within the slab. (Note that due to the slice oversampling implemented in our acquisition, spins just above or below the slab will also be excited.) Thus, small pial veins only contain blood water spins that have experienced a large number of RF pulses due to the long transit time through the pial arterial vasculature, the intracortical arterioles, the capillaries, and the intracortical venules. Accordingly, the blood delivery time for pial veins is much longer as it includes the blood delivery time for pial arteries, in addition to the blood transit time through the entire intracortical vasculature. Hence, their longitudinal magnetization would be similar to that of stationary tissue. To generate an FRE effect in veins, 'pass-through' venous blood from outside the imaging slab is required. This is only available in veins that are passing through the imaging slab, which have much larger diameters. These theoretical considerations are corroborated by the findings in *Figure 9*, where large disconnected vessels with varying intensity profiles were identified as non-arterial. Due to the heterogenous intensity profiles in large veins and the sagittal and transversal sinuses, the intensity-based segmentation applied here may only label a subset of the vessel lumen, creating the impression of many small veins. This is particularly the case for the straight and inferior sagittal sinus in the bottom slab of *Figure 9*. Nevertheless,

future studies potentially combing anatomical prior knowledge, advanced segmentation algorithms, and susceptibility measures would be capable of removing these unwanted veins in post-processing to enable an efficient TOF-MRA image acquisition dedicated to optimally detecting small arteries without the need for additional venous suppression RF pulses.

Once these challenges for vessel segmentation algorithms are addressed, a thorough quantification of the arterial vasculature can be performed. For example, the skeletonization procedure used to estimate the increase of the total length of the segmented vasculature (*Figure 5—figure supplement 1*) exhibits errors particularly in the unwanted sinuses and large veins. While they are consistently present across voxel sizes, and thus may have less impact on relative change in skeleton length, they need to be addressed when estimating the absolute length of the vasculature, or other higher-order features such as number of new branches. (Note that we have also performed the skeletonization procedure on the maximum intensity projections to reduce the number of artefacts and obtained comparable results: reducing the voxel size from 0.8 to 0.5 mm isotropic increases the skeleton length by 44% (3D) vs. 37% (2D), reducing the voxel size from 0.5 to 0.4 mm isotropic increases the skeleton length by 28% (3D) vs. 26% (2D), reducing the voxel size from 0.4 to 0.3 mm isotropic increases the skeleton length by 31% (3D) vs. 16% (2D), and reducing the voxel size from 0.16 to 0.14 mm isotropic increases the skeleton length by 14% (3D) vs. 24% (2D).)

Computational simulations of blood flow based on these segmentations (*Hsu et al., 2017*; *Park et al., 2020*) also require accurate estimates of vessel diameter. However, when vessel diameter and voxel size are similar, diameter estimates often result in large relative errors (*Hoogeveen et al., 1998*; *Klepaczko et al., 2016*). One could imagine using the partial-volume model presented here (*Equation (4)*) to correct for these partial-volume effects (*Nowinski et al., 2011*) by modelling voxel intensities as a mixture of vessel and background intensities. However, this approach requires accurate knowledge of the blood delivery time to estimate the blood signal (*Equation (2)*) and further neglects the fact that more than one tissue class typically surrounds pial vessels (see section *Anatomical architecture of the pial arterial vasculature* and *Figure 1*). In particular the different longitudinal relaxation times of grey matter and cerebrospinal fluid can generate distinct background intensities (*Rooney et al., 2007*; *Wright et al., 2008*). Additionally, the unknown position of the vessel within the voxel plays a significant role when determining the tissue and blood volume fractions, especially when voxel size and vessel diameter are similar. In summary, advanced segmentation algorithms are needed to fully extract the information contained in these data, the removal of unwanted veins might be possible using an additional two-echo TOF acquisition, and vessel diameter estimates need to address potential large errors for vessels that are of similar diameter as the voxel size.

## Future directions

The advantages of imaging the pial arterial vasculature using TOF-MRA without an exogenous contrast agent lie in its non-invasiveness and the potential to combine these data with various other structural and functional image contrasts provided by MRI. One common application is to acquire a velocity-encoded contrast such as phase-contrast MRA (*Arts et al., 2021*; *Bouvy et al., 2016*). Another interesting approach utilizes the inherent TOF contrast in magnetization-prepared two rapid acquisition gradient echo (MP2RAGE) images acquired at ultra-high field that simultaneously acquires vasculature and structural data, albeit at lower achievable resolution and lower FRE compared to the TOF-MRA data in our study (*Choi et al., 2020*). In summary, we expect high-resolution TOF-MRA to be applicable also for group studies to address numerous questions regarding the relationship of arterial topology and morphometry to the anatomical and functional organization of the brain, and the influence of arterial topology and morphometry on brain hemodynamics in humans. Notably, we have focused on imaging pial arteries of the human cerebrum; however, other brain structures such as the cerebellum, subcortex, and white matter are of course also of interest. While the same theoretical considerations apply, imaging the arterial vasculature in these structures will require even smaller voxel sizes due to their smaller arterial diameters (*Duvernoy et al., 1983*; *Duvernoy et al., 1981*; *Nonaka et al., 2003a*). In addition, imaging of the pial *venous* vasculature—using susceptibility-based contrasts such as $T_2$\*-weighted magnitude (*Gulban et al., 2021*) or phase imaging (*Fan et al., 2015*), SWI (*Eckstein et al., 2021*; *Reichenbach et al., 1997*), or quantitative susceptibility mapping (QSM) (*Bernier et al., 2018*; *Huck et al., 2019*; *Mattern et al., 2019*; *Ward et al., 2018*)—would enable

a comprehensive assessment of the complete cortical vasculature and how both arteries and veins shape brain hemodynamics.

When applying measures of arterial topology and morphometry to address questions about the robustness of blood supply (*Baumbach and Heistad, 1985*), arterial territories (*Mut et al., 2014*), or the relationship between the cortical folding pattern and the pial vasculature (*Ii et al., 2020*), particular care needs to be exercised to account for the 'voxel size' bias. By this we mean that the dependency of the vessel contrast on the voxel size shown in our study in conjunction with the reduction in vessel diameter along the vascular tree (*Duvernoy et al., 1981*; *Helthuis et al., 2019*) can introduce a systematic detection bias that varies regionally. At lower imaging resolutions, we would expect particularly high numbers of false negatives, for example, in anterior and posterior brain regions, which are the end points of the pial vascular tree and whose arteries generally have lower vessel diameters. This bias can lead to misinterpretations of the data such as an apparent reduction in vessel density, or an incorrect inference of slower blood flow in these regions. Further, anatomical modelling and vascular network synthesis methods rely on a detailed and unbiased description of the cerebral arterial vasculature to serve as a reference for the developed algorithms (*Bui et al., 2010*; *Ii et al., 2020*; *Keelan et al., 2019*). Finally, computational models of the pial arterial vasculature might be particularly relevant for future investigations into how the local vasculature shapes the spatio-temporal physiological and hemodynamic responses observed in BOLD and non-BOLD fMRI (*Amemiya et al., 2020*; *Bright et al., 2020*; *Chen et al., 2020*; *Drew et al., 2020*).

Estimating the sensitivity of TOF-MRA to pial vessels and verifying our finding that the voxel size is the determining factor for vessel depiction would arguably benefit from complementary imaging methods such as optical coherence tomography, which provides superior sensitivity and specificity for small pial vessel (*Baran and Wang, 2016*). These techniques are commonly used in rodents, but because of their limited penetration depth however, not in humans. Notably, blood velocity measurements can also be obtained (*Wang and An, 2009*; *You et al., 2014*) allowing a direct estimate of blood delivery time to delineate the impact of voxel size and blood delivery time on FRE (*Figure 4*).

In summary, when imaging the pial arterial vasculature we found that—unexpectedly—the limiting factor is not the vascular 'physiology', that is, not slow blood flow in pial arteries, but rather the currently available image resolution. This new perspective provides further motivation to push to higher imaging resolutions, which will provide an even more detailed picture of the pial arterial network. While many challenges remain to obtain high-quality high-resolution images and vascular segmentations, we expect the techniques presented here to allow new insights into the topology and morphometry of the pial arterial vasculature. Given the stark improvements in vessel contrast when using smaller voxels, we believe that as imaging technology progresses, faster and higher-resolved acquisition techniques will become available. Utilizing the potential of higher field strengths, advanced receive coil arrays, new reconstruction techniques, faster imaging gradients, and sophisticated motion correction will enable broad applications of this approach to both basic neuroscience and clinical research.

## Materials and methods
### Simulations
The simulations of the FRE as a function of sequence parameter (*Figure 4*, *Equations 1–3*) or voxel size and blood delivery time (*Figure 5*, *Equations 4–6*) were performed in Matlab R2020a (The MathWorks, Natick, MA) and are available online: https://gitlab.com/SaskiaB/pialvesseltof.git (branch: paper-version). The calculation of the relative blood volume (*Equation (4)*) to estimate the partial-volume FRE (*Equation (6)*) is outlined in the supplementary material (*Estimation of vessel-volume fraction*). Where not stated otherwise, simulations were performed assuming a pial artery diameter of 200 μm, a $T_R$ value of 20 ms, an excitation flip angle of 18°, a blood delivery time of 400 ms, and longitudinal relaxation times of blood and tissue of 2100 and 1950 ms at 7 T, respectively (*Huber, 2014*, Table 2.1).

### Data acquisition
Four healthy adults volunteered to participate in the study (four males, ages 30–46). Prior to imaging, written informed consent was obtained from the three participants scanned in Boston

(*Figures 5, 6, 8 and 9* and corresponding figure supplements) in accordance with the Partners Human Research Committee and the Massachusetts General Hospital Institutional Review Board (protocol #2016P000274); after the study completion, a consent form addendum was used to obtain informed consent from each participant specifically to share their anonymized data on a public data repository. For the single subject from Magdeburg (*Figure 7* and corresponding figure supplements), the consent to share openly the data in anonymized form was acquired prospectively (facultative option in study consent form) in accordance with the 'Ethikkommission Otto-von-Guericke-Universität Magdeburg' (protocol 15/20). Experiments were conducted at 7 T on a Siemens MAGNETOM whole-body scanner (Siemens Healthcare, Erlangen, Germany) equipped with SC72 gradients. Imaging data presented in *Figures 5, 6, 8 and 9* and corresponding figure supplements were acquired at the Athinoula A Martinos Center for Biomedical Imaging, Massachusetts General Hospital, Boston, MA, using an in-house built head-only birdcage volume transmit coil and 31-channel receive coil array. The imaging data presented in *Figure 7* and corresponding figure supplements were acquired at the Institute of Experimental Physics, Otto-von-Guericke-University, Magdeburg, using a quadrature transmit and 32-channel receive head coil (Nova Medical, Wilmington, MA) and an optical, marker-based tracking system with an in-bore camera (Metria Innovation, Milwaukee, WI) for prospective motion correction.

To empirically assess the effect of resolution on FRE, the vendor-supplied TOF sequence was utilized to acquire MRA data at 0.3, 0.4, 0.5, and 0.8 mm isotropic resolution. For each resolution, all other sequence parameters were kept as identical as possible: $T_E$ = 4.73 ms with asymmetric echo, $T_R$ = 20 ms, flip angle = 18°, FOV = 204 mm × 178.5 mm, slab thickness = [21.6, 19.2, 22, 25.6 mm], slice oversampling = [22%, 25%, 18.2%, 12.5%], $R$>>$L$ phase encoding direction, readout bandwidth = 120 Hz/px, GRAPPA (*Griswold et al., 2002*) = 3, number of reference lines = 32, no partial Fourier, 3D centric reordering of the phase encoding steps (*Korin et al., 1992*), read and slab-select flow compensation (*Parker et al., 2003*) resulting in a total acquisition time of 6 min 42 s, 3 min 29 s, 2 min 23 s, and 1 min 14 s, respectively. Note that the employed flow compensation only accounts for signal loss due to the readout and slab-select gradients, but not for potential vessel displacements in the phase-encoding directions. Tilt-optimized, non-saturated excitation (TONE) pulses (*Atkinson et al., 1994*), which are commonly employed to obtain a homogenous image contrast (*Atkinson et al., 1994*; *Carr and Carroll, 2012*; *Nägele et al., 1995*), were not used; because these dedicated RF pulses are most effective when blood flow is perpendicular to the imaging slab and moving in the direction of the TONE pulse ramp, for example, in the *H*>>*F* direction. However, in the case of pial arteries, the complex branching pattern of the vasculature on the folded cortex means that no preferential flow direction is present. Thus, the optimal direction of the TONE ramp is undetermined and therefore this technique is not appropriate for imaging the mesoscopic pial arteries.

The high-resolution TOF acquisitions used the same vendor-supplied TOF sequence, but with one slight modification to allow for larger image encoding matrices. The parameters for the TOF acquisition at 0.16 mm isotropic resolution (*Figure 6* and figure supplements) were the following: $T_E$ = 6.56 ms with asymmetric echo, $T_R$ = 20 ms, flip angle = 18°, FOV = 204 mm × 173.8 mm, slab thickness = 8.32 mm, slice oversampling = 15.4%, $R$>>$L$ phase encoding direction, readout bandwidth = 100 Hz/px, GRAPPA = 2, number of reference lines = 32, no partial Fourier, 3D centric reordering of the phase encoding steps, flow compensation in the read and slab-select directions, resulting in a total acquisition time of 11 min 42 s per slab. Immediately after each scan, the resulting images were visually inspected for motion artefacts, which manifest as a loss of image sharpness, and feedback was provided to the participant about their performance. If motion was detected, the scan was repeated. To cover a larger area, the imaging slab was moved head-foot following each scan and the acquisition was repeated after ensuring that the participant was still comfortable.

The high-resolution TOF acquisition at 0.14 mm isotropic resolution (*Figure 7* and figure supplements) utilized a prospective motion correction system to enable longer scan times and minimize the effect of subject motion. We followed the procedure described in *Mattern et al., 2018*. In brief, a 15 mm × 15 mm marker with a Morié pattern was attached via an individually made mouthpiece to the subject's teeth, skull, and thus brain (*Callaghan et al., 2015*). A camera, which tracked the marker at 80 frames per second, was positioned above the subject's head in the scanner. The rigid-body motion parameters estimated from the video stream were sent to the MRI scanner to update the imaging volume every $T_R$. In total, three imaging slabs were acquired in 1 hr 5 min 40 s covering 19.6 mm in the head-foot direction. The following pulse sequence parameters were used: $T_E$ = 6.99 ms with

asymmetric echo, $T_R$ = 20 ms, flip angle = 18°, FOV = 204 mm × 153 mm, slab thickness = 7.28 mm, slice oversampling = 15.4%, $R>>L$ phase encoding direction, readout bandwidth = 100 Hz/px, no parallel imaging, no partial Fourier, 3D centric reordering of the phase encoding steps, read and slab-select flow compensation resulting in a total acquisition time of 21 min 53 s per slab. For comparison, a single slab at 0.16 mm isotropic resolution with the same pulse sequence parameters described in the previous paragraph was acquired.

To assess the magnitude of the vessel displacement artefact (*Figure 8*) and verify the feasibility of unwanted vein removal during post-processing (*Figure 9*; *Deistung et al., 2009*; *Du and Jin, 2008*), we acquired additional two-echo TOF data in a fourth participant with the following parameters: isotropic voxel size = 0.4 mm, $T_E$ = [7.05 ms, 14 ms] with asymmetric echo, $T_R$ = 20 ms, flip angle = 18°, FOV = 204 mm × 153 mm, slab thickness = 20.8 mm, number of slabs = 4, slice oversampling = 23.1%, $R>>L$ phase encoding direction, readout bandwidth = 200 Hz/px, monopolar readout, GRAPPA = 4, no partial Fourier, no 3D centric reordering of the phase encoding steps, full gradient-moment-based flow compensation in read and phase encoding directions resulting in a total acquisition time of 10 min 42 s.

The anonymized imaging data presented in this manuscript are stored in OSF (Center for Open Science, Inc, Charlottesville, VA) accessible via https://osf.io/nr6gc/. Note that additional multi-contrast high-resolution imaging data are available from this participant (*Lüsebrink et al., 2021*; *Lüsebrink et al., 2017*). Within the OSF repository, high-resolution versions of the figures contained in this manuscript are also provided.

## Data analysis

All imaging data were slab-wise bias-field corrected using the *N4BiasFieldCorrection* (*Tustison et al., 2010*) tool in ANTs (*Avants et al., 2009*) with the default parameters. To compare the empirical FRE across the four different resolutions (*Figure 5*), manual masks were first created for the smallest part of the vessel in the image with the highest resolution and for the largest part of the vessel in the image with the lowest resolution. Then, rigid-body transformation parameters from the low-resolution to the high-resolution (and the high-resolution to the low-resolution) image were estimated using *coregister* in SPM (https://www.fil.ion.ucl.ac.uk/spm/), and their inverse was applied to the vessel mask using SPM's *reslice*. To calculate the empirical FRE (*Equation (3)*), the mean of the intensity values within the vessel mask was used to approximate the blood magnetization, and the mean of the intensity values 1 voxel outside of the vessel mask was used as the tissue magnetization.

For vessel segmentation, a semi-automatic segmentation pipeline was implemented in Matlab R2020a (The MathWorks, Natick, MA) using the UniQC toolbox (*Frässle et al., 2021*): First, a brain mask was created through thresholding which was then manually corrected in ITK-SNAP (http://www.itksnap.org/) (*Yushkevich et al., 2006*) such that pial vessels were included. For the high-resolution TOF data (*Figures 6 and 7* and figure supplements), denoising to remove high-frequency noise was performed using the implementation of an adaptive non-local means denoising algorithm (*Manjón, 2010*) provided in *DenoiseImage* within the ANTs toolbox, with the search radius for the denoising set to 5 voxels and noise type set to Rician. Next, the brain mask was applied to the bias corrected and denoised data (if applicable). Then, a vessel mask was created based on a manually defined threshold, and clusters with less than 10 or 5 voxels for the high- and low-resolution acquisitions, respectively, were removed from the vessel mask. Finally, an iterative region-growing procedure starting at each voxel of the initial vessel mask was applied that successively included additional voxels into the vessel mask if they were connected to a voxel which was already included and above a manually defined threshold (slightly below the previous threshold). Both thresholds were applied globally but manually adjusted for each slab. No correction for motion between slabs was applied. The Matlab code describing the segmentation algorithm as well as the analysis of the two-echo TOF acquisition outlined in the following paragraph are also included in our GitHub repository (https://gitlab.com/SaskiaB/pialvesseltof.git, *Bollmann, 2022*; copy archived at swh:1:rev:4b23440d98f8d02b0ecb-8f839afc3b6c81c90969). To estimate the increased detection of vessels with higher resolutions, we computed the relative increase in the length of the segmented vessels for the data presented in *Figure 5* (0.8, 0.5, 0.4, and 0.3 mm isotropic voxel size) and *Figure 7* (0.16 and 0.14 mm isotropic voxel size) by computing the skeleton using the *bwskel* Matlab function and then calculating the skeleton length as the number of voxels in the skeleton multiplied by the voxel size.

The stronger background signal in the lower-resolution two-echo TOF acquisition necessitated a more advanced vessel segmentation algorithm. We therefore utilized the procedure described in *Bernier et al., 2018*, to derive the initial vessel mask, before performing the region-growing procedure. To illustrate the vessel displacement, segmentation of the second echo was skeletonized using the *bwskel* Matlab function, which was then overlaid on the MIP of the first echo (*Figure 8*) to highlight the vessel centreline shift between the two echoes. The voxel-wise $T_2{}^*$ values for the vein removal were estimated from the natural logarithm of the intensity ratios of the two echoes and the echo time difference. Vessels were deemed to be unwanted veins if the 90% percentile of the $T_2{}^*$ values in each independent (unconnected) vessel/vessel tree was below 19 ms (*Figure 9*) or 27 ms (*Figure 7*).

To estimate the vessel density mentioned in the *Anatomical architecture of the pial arterial vasculature* section, the drawing in Figure 239 in *Duvernoy, 1999*, was segmented into arteries, veins, and background using their RGB values to estimate their relative content.

## Acknowledgements

We thank Mr Kyle Droppa and Ms Nina Fultz for help with subject recruitment and MRI scanner operation. This work was supported in part by the NIH NIBIB (grants P41-EB015896, P41-EB030006, R01-EB032746, and R01-EB019437), NINDS (grant R21-NS106706), NIMH (grant R01-MH124004), NIA (grant RF1-AG074008), the *BRAIN Initiative* (NIH NIMH grants R01-MH111438 and R01-MH111419 and NIH NINDS grant U19-NS123717), the Natural Sciences and Engineering Research Council of Canada (NSERC), the *Fonds de recherche Nature et technologies* (FQRNT), the German Research Foundation (DFG) (MA 9235/1–1), and the MGH/HST Athinoula A Martinos Center for Biomedical Imaging; and was made possible by the resources provided by NIH Shared Instrumentation Grants S10-RR019371 and S10-OD02363701. SRR was supported by the Marie Skłodowska-Curie Action MS-fMRI-QSM 794298.

## Additional information

### Funding

| Funder | Grant reference number | Author |
|---|---|---|
| National Institute of Biomedical Imaging and Bioengineering | P41-EB015896 | Jonathan R Polimeni |
| National Institute of Biomedical Imaging and Bioengineering | P41-EB030006 | Jonathan R Polimeni |
| National Institute of Biomedical Imaging and Bioengineering | R01-EB019437 | Jonathan R Polimeni |
| National Institute of Neurological Disorders and Stroke | R21-NS106706 | Jonathan R Polimeni |
| National Institute of Mental Health | R01-MH111438 | Jonathan R Polimeni |
| National Institute of Mental Health | R01-MH111419 | Saskia Bollmann Jonathan R Polimeni Michaël Bernier |
| Natural Sciences and Engineering Research Council of Canada | | Michaël Bernier |
| Fonds de recherche du Québec – Nature et technologies | | Michaël Bernier |

| Funder | Grant reference number | Author |
|---|---|---|
| Deutsche Forschungsgemeinschaft | MA 9235/1-1 | Hendrik Mattern Oliver Speck |
| National Institutes of Health | S10-RR019371 | Jonathan R Polimeni |
| National Institutes of Health | S10-OD02363701 | Jonathan R Polimeni |
| European Commission | MS-fMRI-QSM 794298 | Simon D Robinson |
| National Institute on Aging | RF1-AG074008 | Jonathan R Polimeni |
| National Institute of Neurological Disorders and Stroke | U19-NS123717 | Jonathan R Polimeni |
| National Institute of Mental Health | R01-MH124004 | Jonathan R Polimeni |
| National Institute of Biomedical Imaging and Bioengineering | R01-EB032746 | Jonathan R Polimeni |

The funders had no role in study design, data collection and interpretation, or the decision to submit the work for publication.

## Author contributions

Saskia Bollmann, Conceptualization, Data curation, Formal analysis, Investigation, Methodology, Project administration, Software, Visualization, Writing - original draft, Writing – review and editing; Hendrik Mattern, Formal analysis, Investigation, Methodology, Software, Writing – review and editing; Michaël Bernier, Conceptualization, Software, Writing – review and editing; Simon D Robinson, Daniel Park, Resources, Software, Writing – review and editing; Oliver Speck, Funding acquisition, Methodology, Project administration, Resources, Writing – review and editing; Jonathan R Polimeni, Conceptualization, Funding acquisition, Project administration, Resources, Supervision, Writing - original draft

## Author ORCIDs

Saskia Bollmann http://orcid.org/0000-0001-8242-8008
Hendrik Mattern http://orcid.org/0000-0001-5740-4522
Daniel Park http://orcid.org/0000-0002-7263-9327
Jonathan R Polimeni http://orcid.org/0000-0002-1348-1179

## Ethics

Four healthy adults volunteered to participate in the study (four males, ages 30-46). Prior to imaging, written informed consent was obtained from the three participants scanned in Boston (Figure 5, 6, 8 and 9 and corresponding figure supplements) in accordance with the Partners Human Research Committee and the Massachusetts General Hospital Institutional Review Board (protocol #2016P000274); after the study completion, a consent form addendum was used to obtain informed consent from each participant specifically to share their anonymized data on a public data repository. For the single subject from Magdeburg (Figure 7 and corresponding figure supplements) the consent to share openly the data in anonymized form was acquired prospectively (facultative option in study consent form) in accordance with the 'Ethikkommission Otto-von-Guericke-Universität Magdeburg' (protocol 15/20).

## Decision letter and Author response

Decision letter https://doi.org/10.7554/eLife.71186.sa1
Author response https://doi.org/10.7554/eLife.71186.sa2

---

# Additional files

## Supplementary files
• Transparent reporting form

## Data availability

The anonymized imaging data presented in this manuscript are stored in OSF (OSF, Center for Open Science, Inc, Charlottesville, Virginia, USA) accessible via https://doi.org/10.17605/OSF.IO/NR6GC.

The following dataset was generated:

| Author(s) | Year | Dataset title | Dataset URL | Database and Identifier |
|---|---|---|---|---|
| Bollmann S, Mattern H, Bernier M, Robinson SD, Park D, Speck O, Polimeni JR | 2022 | Imaging of the Pial Arterial Vasculature | https://doi.org/10.17605/OSF.IO/NR6GC | Open Science Framework, 10.17605/OSF.IO/NR6GC |

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

# Appendix 1

## Supplementary material

### Estimation of vessel-volume fraction

To estimate the relative blood-volume fraction $V_{\text{blood}}^{\text{rel}} = V_{\text{vesselInVoxel}}/V_{\text{voxel}}$, we assume a centrally located vessel with radius $r_{\text{vessel}}$ in a voxel with equal length $l_{\text{voxel}}$ along each side. The vessel cross-sectional area, orthogonal to the main axis of the vessel and intersecting the voxel, is split into 8 equal areas $A$, as indicated in *Figure 4—figure supplement 1*. The relative blood volume fraction $V_{\text{blood}}^{\text{rel}}$ can then be computed as the sum of these 8 compartments times the voxel length $l_{\text{voxel}}$ relative to the voxel volume:

$$V_{\text{blood}}^{\text{rel}} = \frac{8 \cdot A \cdot l_{\text{voxel}}}{V_{\text{voxel}}}. \tag{8}$$

If $l_{\text{voxel}} < \cos\frac{\pi}{2} \cdot r_{\text{vessel}}$ (case I), that is, the voxel is completely contained in the vessel, the total vessel area in the voxel is

$$A_{\text{case I}} = \frac{1}{2} \cdot \left(\frac{l_{\text{voxel}}}{2}\right)^2. \tag{9}$$

Conversely, if $r_{\text{vessel}} < l_{\text{voxel}}/2$ (case III), that is, the vessel is completely contained in the voxel, the total vessel area in the voxel is

$$A_{\text{case III}} = \frac{1}{2} \cdot r_{\text{vessel}}^2 \cdot \frac{\pi}{4}. \tag{10}$$

In the case when vessel and voxel intersect, the total area $A$ is the sum of the area $A'$ of the right triangle and area $A''$ of the circular sector:

$$A_{\text{case II}} = A' + A''$$

$$A' = \frac{1}{2} \cdot a \cdot \frac{l_{\text{voxel}}}{2}$$

$$a = \sin\theta \cdot r_{\text{vessel}}$$

$$\theta = \cos^{-1}\frac{\frac{l_{\text{voxel}}}{2}}{r_{\text{vessel}}}$$

$$A'' = \frac{1}{2} \cdot r_{\text{vessel}}^2 \cdot \left(\frac{\pi}{4} - \theta\right). \tag{11}$$

### Effect of FRE definition and interaction with partial-volume model

For the definition of the FRE effect in this study, we used a measure of relative FRE (*Al-Kwifi et al., 2002*) in combination with a partial-volume model (*Equation (6)*). To illustrate the effect of these two definitions, as well as their interaction, we have estimated the relative and absolute FRE for an artery with a diameter of 200 and 2000 µm (i.e. no partial-volume effects). The absolute FRE explicitly takes the voxel volume into account, that is, instead of *Equation (6)* for the relative FRE we used

$$\text{FRE}_{\text{PV}}\left(n_{\text{RF}}, l_{\text{voxel}}, d_{\text{vessel}}\right) = M_z^{\text{total}}\left(n_{\text{RF}}, l_{\text{voxel}}, d_{\text{vessel}}\right) \cdot l_{\text{voxel}}^3 - M_{z\text{S}}^{\text{tissue}} \cdot \left(l_{\text{voxel}}^3\right). \tag{12}$$

Note that the division by $M_{z\text{S}}^{\text{tissue}} \cdot l_{\text{voxel}}^3$ to obtain the relative FRE removes the contribution of the total voxel volume $\left(l_{\text{voxel}}^3\right)$.

*Figure 4—figure supplement 2* shows that, when partial-volume effects are present, the highest relative FRE arises in voxels with the same size as or smaller than the vessel diameter (*Figure 4—figure supplement 2A*), whereas the absolute FRE increases with voxel size (*Figure 4—figure supplement 2C*). If no partial-volume effects are present, the relative FRE becomes independent of voxel size (*Figure 4—figure supplement 2B*), whereas the absolute FRE increases with voxel size (*Figure 4—figure supplement 2D*). While the partial-volume effects for the relative FRE are substantial, they are much more subtle when using the absolute FRE and do not alter the overall characteristics.

## Optimization of repetition time and excitation flip angle

As the main goal of the optimization here was to start within an already established parameter range for TOF imaging at ultra-high field (*Kang et al., 2010*; *Stamm et al., 2013*; *Von Morze et al., 2007*), we only needed to then further tailor these for small arteries by considering a third parameter, namely the blood delivery time. From a practical perspective, a $T_R$ of 20 ms as a reference point was favourable, as it offered a time-efficient readout minimizing wait times between excitations but allowing low encoding bandwidths to maximize SNR. Due to the interdependency of flip angle and repetition time, for any *one* blood delivery time any FRE could (in theory) be achieved. For example, a similar FRE curve at 18° flip angle and 5 ms $T_R$ can also be achieved at 28° flip angle and 20 ms $T_R$; or the FRE curve at 18° flip angle and 30 ms $T_R$ is comparable to the FRE curve at 8° flip angle and 5 ms $T_R$ (*Figure 3—figure supplement 1*, top). In addition, the difference between optimal parameter settings diminishes for long blood delivery times, such that at a blood delivery time of 500 ms (*Figure 3—figure supplement 1*, bottom), the optimal flip angle at a $T_R$ of 15, 20, or 25 ms would be 14°, 16°, and 18°, respectively. This is in contrast to a blood delivery time of 100 ms, where the optimal flip angles would be 32°, 37°, and 41°. In conclusion, in the regime of small arteries, long $T_R$ values in combination with low flip angles ensure FRE at blood delivery times of 200 ms and above, and within this regime there are marginal gains by further optimizing parameter values and the optimal values are all similar.

## Vessel displacement artefact

The origin of the vessel displacement artefact described in section *Velocity- and $T_E$-dependent vessel displacement artefacts* and shown in *Figure 8* is illustrated in *Figure 8—figure supplement 1A* (*Brown et al., 2014b*). In short, the moving blood water spins accumulate the phase increment at time $t_{pe}$ at position ($x_{tpe}$, $y_{tpe}$) encoding their position in the phase encoding direction $y$, but continue to flow along the blood vessel such that at the echo time ($T_E$) they are in position ($x_{tTE}$, $y_{tTE}$). Consequently, during image reconstruction the signal from these spins is located in position ($x_{tTE}$, $y_{tpe}$) which they have never occupied. The magnitude of the vessel displacement is simply the distance travelled by the blood water spins in the time between the phase-encoding blip and the echo time, and can be described as in *Equation (7)* assuming a constant blood velocity. *Figure 8—figure supplement 1B* illustrates that an inversion of the polarity of the phase encoding direction, a technique commonly used to estimate and correct for voxel displacements found in echo-planar imaging stemming from main magnetic field inhomogeneities (*Andersson et al., 2003*; *Jezzard and Clare, 1999*), does *not* change the direction of the displacement. This is because the vessel displacement is caused by true object motion, and while an inversion of the polarities would (in this example) imprint the 'correct' larger phase increment on the blood water spins from Panel A, this phase increment is now also associated with the inverted position and thus the vessel displacement occurs in the same direction.

## Potential for advanced segmentations algorithms

The potential for advanced segmentation algorithms is illustrated in *Figure 7—figure supplement 3*, where a small section of the TOF data acquired at 0.14 mm isotropic resolution from *Figure 7* was manually segmented. In this example, it is clear that the automatic segmentation is missing several small vessel branches that could be readily segmented manually. The maximum intensity projection contains 50 slices, corresponding to 7 mm coverage in the head-foot direction. Improved performance of this manual segmentation generated on a per slice basis using the full 3D information shows the potential for advances in automatic segmentation algorithms to identify many more pial arteries in this data than which are visible in the maximum intensity projections presented here.

