## [Editor Report]

This article revisits a classic magnetic resonance imaging technique to image brain vasculature, showing that small blood vessels, deemed too difficult to image, could be targeted effectively with extremely high-resolution imaging. This proof-of-concept, both theoretical and experimental, may be of use for building detailed models of brain physiology and detecting fine alterations of blood vessels in disease.

---

## [Decision Letter]

**Decision letter after peer review:**

Thank you for submitting your article "Imaging of the pial arterial vasculature of the human brain in vivo using high-resolution 7T time-of-flight angiography" for consideration by *eLife*. Your article has been reviewed by 2 peer reviewers, and the evaluation has been overseen by a Reviewing Editor and Chris Baker as the Senior Editor. The reviewers have opted to remain anonymous.

Essential revisions:

In addition to commenting on the other points raised, please pay special attention to the below essential revisions:

1 – More quantitative analyses.

2 – Introduce TOF-MRA for the benefit of the broader *eLife* readership.

3 – Consider the effect of noise in some of the analyses.

---

## [Author Response]

Essential revisions:In addition to commenting on the other points raised, please pay special attention to the below essential revisions:1 – More quantitative analyses.2 – Introduce TOF-MRA for the benefit of the broader eLife readership.3 – Consider the effect of noise in some of the analyses.

We would like to thank the reviewers for their valuable and engaging comments. We have addressed all the points raised and revised the manuscript accordingly.